# State-Constrained Offline Reinforcement Learning

**Charles A. Hepburn**[1,3], **Yue Jin**[2], **Giovanni Montana**[2,3,4]

[1] *Mathematics Institute, University of Warwick, Coventry, UK*

[2] *Warwick Manufacturing Group, University of Warwick, Coventry, UK*

[3] *Department of Statistics, University of Warwick, Coventry, UK*

[4] *Alan Turing Institute, London, UK*

*{charlie.hepburn, yue.jin.3, g.montana}@warwick.ac.uk*

**Reviewed on OpenReview:** *https://openreview.net/forum?id=KcR8ykFlHA*

## Abstract

Traditional offline reinforcement learning (RL) methods predominantly operate in a batch-constrained setting. This confines the algorithms to a specific state-action distribution present in the dataset, reducing the effects of distributional shift but restricting the policy to seen actions. In this paper, we alleviate this limitation by introducing *state-constrained* offline RL, a novel framework that focuses solely on the dataset's state distribution. This approach allows the policy to take high-quality out-of-distribution actions that lead to in-distribution states, significantly enhancing learning potential. The proposed setting not only broadens the learning horizon but also improves the ability to combine different trajectories from the dataset effectively, a desirable property inherent in offline RL. Our research is underpinned by theoretical findings that pave the way for subsequent advancements in this area. Additionally, we introduce StaCQ, a deep learning algorithm that achieves state-of-the-art performance on the D4RL benchmark datasets and aligns with our theoretical propositions. StaCQ establishes a strong baseline for forthcoming explorations in this domain.

## 1 Introduction

Offline RL aims to derive an optimal policy solely from a fixed dataset of pre-collected experiences, without further interactions with the environment (Lange et al., 2012; Levine et al., 2020). It prohibits online exploration, meaning that effective policies must be constructed solely from the evidence provided by an unknown, potentially sub-optimal behaviour policy active in the environment. This approach is especially suited for real-world scenarios where executing sub-optimal actions can be dangerous, time-consuming, or costly, yet there is an abundance of prior data. Relevant applications include robotics (Singh et al., 2020; Kumar et al., 2021; Sinha et al., 2022), long-term healthcare treatment plans (Tang & Wiens, 2021; Tang et al., 2022; Shiranthika et al., 2022), and autonomous driving (Shi et al., 2021; Fang et al., 2022; Diehl et al., 2023). Despite its practical appeal, offline RL faces a significant distributional shift challenge (Kumar et al., 2019). This phenomenon arises when attempting to estimate the values of actions not present in the dataset, often manifesting as an overestimation. As a result, out-of-distribution (OOD) actions are perceived as more valuable than they truly are, causing the agent to select sub-optimal actions and leading to the accumulation of errors (Fujimoto et al., 2019).

Currently available offline RL techniques mitigate the issue of distributional shift through two primary strategies. The first constrains the policy, during training, to take actions close to dataset actions (Fujimoto et al., 2019; Kumar et al., 2019; Wu et al., 2019; Siegel et al., 2020; Kostrikov et al., 2021a; Zhou et al., 2021; Fujimoto & Gu, 2021). The second revolves around conservatively estimating the value of OOD actions (Kumar et al., 2020; Yu et al., 2021; An et al., 2021). Both strategies aim to anchor the learning algorithm to the state-action pairs within the dataset, an approach termed batch-constrained (Fujimoto et al., 2019). By doing so, these methods seek to minimise the impact of distributional shift. However, adhering strictly to the

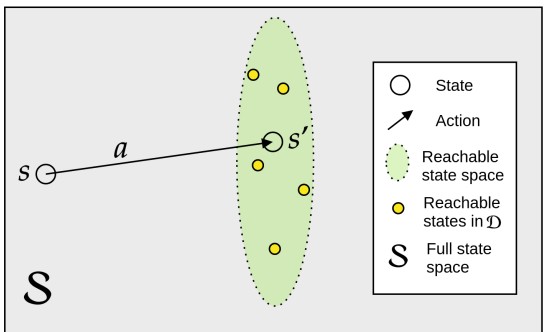

Figure 1: Illustration of state reachability in a continuous state space. States ($s$) and next states ($s'$) from the dataset are shown, with yellow circles representing additional reachable states. The state-constrained method identifies high-quality reachable states to mitigate distributional shift while improving policy performance.

state-action distribution can be restrictive when the optimal action is not found in the dataset. Thus, recent techniques seek a balance between exploiting potentially valuable OOD actions and minimising distributional shift repercussions.

Given the constraints of the state-action distribution, a pivotal question arises: *can we confine our methods to the state distribution alone and still counteract distributional shift?*

Constraining exclusively to the dataset's state distribution could significantly reduce the requisite dataset size. Instead of needing all state-action pairs, the focus shifts to states alone. This approach allows the agent to initiate OOD actions, provided they lead to a known state, a form of safe action exploration which avoids distributional shift. Such flexibility can enhance the capability of offline RL algorithms in trajectory stitching — combining sub-optimal experiences into improved trajectories. Rather than connecting based on actions from different trajectories, the state-constrained approach facilitates stitching by leveraging adjacent states. This method relies on understanding these proximate states, a concept we label as *reachability*, which refers to the ability to reach a particular state from another state within the dataset.

This idea is illustrated in Figure 1, which shows the benefits of a state-constrained methodology. Most current offline RL methods constrain to the specific transition, $(s, a)$ or $(s, s')$-pair, found in the dataset. Our state-constrained methodology can instead find alternative next states (yellow circles in Figure 1) and select the highest-value one. This approach effectively increases the number of transitions for the agent to learn from. It avoids distributional shift by updating on in-distribution states and leads to a higher-quality policy, as the policy is no longer restricted to explicitly observed actions.

In the state-constrained approach, understanding reachability is crucial for effectively stitching together sub-optimal trajectories and improving the overall policy. In some environments, reachability is easily delineated; for example, in grid-based environments like mazes, where the agent can move directly between adjacent cells. In other environments, reachability must be ascertained from the dataset, such as in complex robotic systems where state transitions depend on intricate dynamics.

In this work, we make several key contributions. First, building on the concept of reachability, we introduce state-constrained offline RL. Unlike batch-constrained RL, which anchors learning to specific state–action pairs, state-constrained RL focuses on states, allowing more flexible action selection while still mitigating distributional shift. We also provide theoretical convergence guarantees under deterministic assumptions, showing that our framework yields policies whose actions are of higher or equal value compared to batch-constrained approaches. Another main contribution is StaCQ, a novel deep learning algorithm that learns a state-constrained value function and updates the policy to stay close to the highest-quality reachable states. We demonstrate competitive performance on multiple D4RL tasks, surpassing many state-of-the-art methods in locomotion and Antmaze. This positions StaCQ as a robust baseline for future work on state-constrained

RL, similarly to how BCQ (Fujimoto et al., 2019) has served as a strong baseline for batch-constrained techniques, thereby advancing offline RL research.

## 2   Preliminaries

The RL framework involves an agent interacting with an environment, which is typically represented as a Markov Decision Process (MDP). Such an MDP is defined as $\mathcal{M} = (\mathcal{S}, \mathcal{A}, P, R, \gamma)$, where $\mathcal{S}$ and $\mathcal{A}$ denote the state and action spaces, respectively; $P = p(s'|s, a)$ represents the environment transition dynamics; $R = r(s, a, s')$ is the reward function for transitioning; and $0 \leq \gamma < 1$ is the discount factor (Sutton & Barto, 2018). As in Fujimoto et al. (2019), we focus on the deterministic MDP, where $p(s'|s, a) = \{0, 1\}$ and $r(s, a, s') = r(s, s')$. In RL, the agent's objective is to identify an optimal policy, $\pi(s)$, that maximises the future discounted sum of rewards $\sum_{i=t} \gamma^{i-t} r_i(s_i, s_i')$.

QSA-values, $Q(s, a) = \mathbb{E}_\pi[\sum_{i=0} \gamma^t r_t(s_t, a_t)|s_0 = s, a_0 = a]$, are the expected sum of future discounted rewards for executing action $a$ in state $s$ and there after following policy $\pi$. Q-learning (Watkins & Dayan, 1992), denoted as QSA-learning in this paper, estimates the QSA-values under the assumption that future actions are chosen optimally,

$$Q(s, a) \leftarrow (1 - \alpha)Q(s, a) + \alpha\Big[r(s, a, s') + \gamma \max_{a'} Q(s', a')\Big]. \tag{1}$$

The optimal policy derived from QSA-learning identifies the best action that maximises the QSA-values, $=\pi^*(s) = \arg\max_a Q(s, a)$. In deterministic contexts, QSA-learning parallels QSS-learning (Edwards et al., 2020), which estimates the expected rewards upon transitioning from state $s$ to $s'$, followed by optimal decisions:

$$Q(s, s') \leftarrow (1 - \alpha)Q(s, s') + \alpha\Big[r(s, s') + \gamma \max_{s''} Q(s', s'')\Big]. \tag{2}$$

QSS-learning enables QSS-values to be learned for transitioning, without evaluating actions. The optimal policy via QSS-learning discerns the most advantageous subsequent state to maximise QSS-values, $\pi_s^*(s) = \arg\max_{s'} Q(s, s')$. An action can then be retrieved from an inverse dynamics model as $a = I(s, \pi_s^*(s))$, a strategy originally designed to address QSA-learning's challenges in redundant action spaces, i.e. where multiple actions lead to the same next state. The methods presented in this paper build upon QSS-learning, using it to avoid evaluating OOD actions between reachable state pairs.

In offline RL, the agent aims to discover an optimal policy; however, it must solely rely on a fixed dataset without further interactions with the environment (Lange et al., 2012; Levine et al., 2020). The dataset comprises trajectories made up of transitions consisting of the current state, action, next state, and the reward for transitioning, where actions are chosen based on an unknown behaviour policy, $\pi_\beta$. Batch-Constrained QSA-learning (BCQL) (Fujimoto et al., 2019) adapts QSA-learning for the offline setting by restricting the optimisation to state-action pairs present in the dataset:

$$Q(s, a) \leftarrow (1 - \alpha)Q(s, a) + \alpha\Big[r(s, a, s') + \gamma \max_{a' \text{ s.t.} (s', a') \in \mathcal{D}} Q(s', a')\Big]. \tag{3}$$

This approach is restrictive since convergence to the optimal value function in all states is only ensured if every optimal $(s, a)$-pair from the MDP resides within the dataset. Such a limitation aims to sidestep extrapolation errors resulting from distributional shifts. Nonetheless, this framework places a significant constraint on any learning algorithm. Subsequent sections relax this constraint: instead of adhering strictly to state-action pairs (batch-constrained), the learning process is only bound by states (state-constrained). In the following section, we introduce the state-constrained framework and provide a new algorithm called state-constrained QSS-learning (SCQL). SCQL is based from QSS-learning which is useful to evaluate state and reachable next state pairs, where both states exist in the dataset, avoiding the distributional shift issue in offline RL. We show that, under minor assumptions, SCQL converges to the optimal QSS-value and produces a less-restrictive policy than BCQL.

## 3 State-constrained QSS-learning

In this section, we provide a formal introduction to state-constrained QSS-learning and establish its convergence to the optimal QSS-value under a set of minimal assumptions. Within this framework, the learning updates are restricted exclusively to the states present in the dataset. A rigorous definition of state reachability is indispensable for this setting:

**Definition 3.1. (State reachability)** In a deterministic MDP $\mathcal{M}$, a state $s'$ is considered reachable from state $s$ if and only if there exists an action $a$ such that $p(s'|s,a) = 1$. We denote the set of states reachable from $s$ as $\mathcal{SR}_{\mathcal{M}}(s)$, where $s' \in \mathcal{SR}_{\mathcal{M}}(s)$.

Definition 3.1 implies that a state is reachable if there exists an action that, when executed in the environment, leads to that state. This definition allows multiple reachable next states to be evaluated rather than a single state-action pair. State-constrained QSS-learning enables more flexible learning updates by evaluating multiple reachable next states, as shown in Figure 1 where the agent can now learn from all reachable states rather than the single explicit next state. This flexibility leads to more robust policies that can better generalise beyond the specific transitions seen in the dataset.

### 3.1 Theoretical foundations

In this section, we initially adapt the theorems presented for BCQL (Fujimoto et al., 2019) to suit the state-constrained context (Theorems 3.4 - 3.6). All our subsequent theorems are proposed based on learning an optimal QSS-value. However, it is important to note that these theorems still hold if QSA-learning is used instead, with an inverse model defined to evaluate actions between states and reachable next states.

Our theory operates under the following assumptions: (A1) the environment is deterministic; (A2) the rewards are bounded such that $\forall(s,s'), |r(s,s')| \leq c$; (A3) the QSS-values, $Q(s,s')$, are initialised to finite values; and (A4) the discount factor is set such that $0 \leq \gamma < 1$.

First, we show that under these assumptions, QSS-learning converges to the optimal QSS-value.

**Theorem 3.2.** *Under assumptions A1-4, and with the training rule given by*

$$Q(s,s') \leftarrow r(s,s') + \gamma \max_{s'' \ s.t. \ s'' \in \mathcal{SR}_{\mathcal{M}}(s')} Q(s',s''), \tag{4}$$

*assuming each $(s,s')$ pair is visited infinitely often, let $Q_n(s,s')$ be the value from the nth update of Eq. (4), then $Q_n(s,s')$ converges to the optimal QSS-value, $Q^*(s,s')$, as $n \to \infty$ for all $s,s'$.*

*Proof.* This follows from the convergence of QSA-learning in a deterministic MDP (Mitchell, 1997). For brevity and clarity, the full proof is given in the Appendix. □

The convergence of QSS-learning in a deterministic MDP, as shown in Theorem 3.2, is crucial for establishing the convergence of SCQL. Since SCQL is based on QSS-learning and operates in a deterministic MDP, Theorem 3.2 provides the foundation for proving the convergence and optimality of SCQL under certain assumptions. We now demonstrate that learning the value function from the dataset $\mathcal{D}$ is equivalent to determining the value function of an associated MDP, denoted as $\mathcal{M}_{\mathcal{S}}$. Intuitively, $\mathcal{M}_{\mathcal{S}}$ is the MDP where all transitions are possible between reachable states found in $\mathcal{D}$.

**Definition 3.3. (State-constrained MDP)** Let the state-constrained MDP $\mathcal{M}_{\mathcal{S}} = (\mathcal{S}, \mathcal{A}, \mathcal{P}_{\mathcal{S}}, \mathcal{R}, \gamma)$. Here, both $\mathcal{S}$ and $\mathcal{A}$ remain identical to those in the original MDP, $\mathcal{M}$ and $s_{\text{terminal}}$ is an additional terminating state. The transition probability is given by: $a = I(s,s') \in \mathcal{A}$

$$p_{\mathcal{S}}(s'|s,a) = \begin{cases} 1 & \text{if } (s,s' \in \mathcal{D} \text{ and } s' \in \mathcal{SR}_{\mathcal{M}}(s)) \text{ or } (s \notin \mathcal{D} \text{ and } s' = s_{\text{terminal}}) \\ 0 & \text{otherwise.} \end{cases}$$

The reward function and discount factor remain the same as the original MDP. Except for the terminal state where $r(s, s_{\text{terminal}})$, is set to the initialised value of $Q(s,s')$.

The transition probability $p_{\mathcal{S}}(s'|s,a)$ in the state-constrained MDP is defined such that transitions are possible only between states that are reachable from one to the other according to the state reachability definition. This ensures that the state-constrained MDP captures the essential dynamics of the original MDP while focusing on the states present in the dataset. The reward $r(s,s')$ is assigned as the original reward defined in $\mathcal{M}$. For the case where $s$ is absent from the dataset, the rewards are set to the initialised values $Q(s,s')$. Importantly, the $s$ and $s'$ in Definition 3.3 both exist in the dataset but may not exist as a pair $(s,s')$; this means that more transitions exist under this definition than in the batch-constrained MDP defined in Fujimoto et al. (2019).

**Theorem 3.4.** *By sampling $s$ from $\mathcal{D}$, sampling $s'$ from $\mathcal{SR}_{\mathcal{M}}(s)$ and performing QSS-learning on all reachable state-next state pairs, QSS-learning converges to the optimal value function of the state-constrained MDP $\mathcal{M}_{\mathcal{S}}$.*

*Proof.* Given in Appendix A. $\qquad\square$

Theorem 3.4 establishes that performing QSS-learning on all reachable state-next state pairs from the dataset converges to the optimal value function of the state-constrained MDP. This result is crucial for understanding the convergence and optimality properties of SCQL, as it shows that QSS-learning effectively learns the optimal value function of the state-constrained MDP, which is closely related to the original MDP.

We are now ready to define the state-constrained QSS-learning (SCQL) update which is similar to the BCQL formulation except now the maximisation is constrained to the states rather than state-action pairs in the dataset. This formulation allows the maximisation to be taken over more values composing more accurate Q-values while still staying close to the dataset:

$$Q(s,s') \leftarrow (1-\alpha)Q(s,s') + \alpha\Big[r(s,s') + \gamma \max_{\substack{s'' \, s.t. s'' \in \mathcal{D} \\ \cap \\ s'' \in \mathcal{SR}_{\mathcal{M}}(s')}} Q(s',s'')\Big]. \tag{5}$$

SCQL, Eq. (5), converges under the identical conditions as traditional QSS-learning, primarily because the state-constrained setting is non-limiting whenever every state in the MDP is observed.

**Theorem 3.5.** *Under assumptions A1-4 and assuming every state $s$ is encountered infinitely, let $Q_n(s,s')$ be the value from the nth update of Eq.(5), the update rule of SCQL, then $Q_n(s,s')$ converges to the optimal QSS-value $Q^*(s,s')$, as $n \to \infty$ for all $(s,s')$.*

*Proof.* This follows from Theorem 3.2, noting the state-constraint is non-restrictive with a dataset which contains all possible states. $\qquad\square$

Theorem 3.5 is a reduction in the restriction compared to BCQL as now we only require every state to be encountered infinitely rather than every $(s,a)$-pair.

The optimal policy for our state-constrained approach can be formulated as:

$$\pi_s^*(s) = \underset{\substack{s' \, s.t. s' \in \mathcal{D} \\ \cap \\ s' \in \mathcal{SR}_{\mathcal{M}}(s)}}{\arg\max} Q^*(s,s'). \tag{6}$$

Here, the maximisation is taken over next states from the dataset and that are reachable from the current state. We now demonstrate that Eq. (6) represents the optimal state-constrained policy.

**Theorem 3.6.** *Under assumptions A1-4 and assuming every state $s$ is encountered infinitely, let $Q_n(s,s')$ be the value from the nth update of Eq.(5), the update rule of SCQL, then $Q_n(s,s')$ converges to $Q_{\mathcal{S}}^\pi(s,s')$, the optimal QSS-value computed from states from $\mathcal{D}$, with the optimal state-constrained policy defined by Eq. (6) where $s \in \mathcal{D}$ and $s' \in \mathcal{SR}_{\mathcal{M}}(s) \cap \mathcal{D}$.*

*Proof.* Given in Appendix A. $\qquad\square$

Theorem 3.6 has important practical implications for the performance of SCQL in real-world scenarios with limited datasets. It suggests that SCQL can learn an optimal state-constrained policy even when the dataset does not contain all possible state-action pairs, as long as every state is visited infinitely often. The state-constrained approach greatly reduces the limitations placed on the learning algorithm compared to the batch-constrained method. In the batch-constrained approach, the necessity for every state-action pair to be present in the dataset demands a dataset of size at least $|\mathcal{S}| \times |\mathcal{A}|$. On the other hand, the state-constrained method only mandates that each state be visited, which minimally requires a dataset size of $|\mathcal{S}|$.

We will now show that BCQL is a special case of SCQL, and that policies produced by SCQL will never be worse than policies produced by BCQL.

**Theorem 3.7.** *Let $\pi_{BCQL}$ and $\pi_{SCQL}$ be the policies produced by BCQL and SCQL respectively. Then, in a deterministic MDP, $V_{\pi_{SCQL}}(s) \geq V_{\pi_{BCQL}}(s)$, $\forall s \in \mathcal{D}$.*

*Proof.* Given in Appendix A. $\square$

From Theorems 3.6 and 3.7, the state-constrained approach poses fewer restrictions than the batch-constrained counterpart while also always producing a policy at least as good. SCQL can perform more Q-value updates and maximise over more states than BCQL because it considers all reachable state-next state pairs from the dataset, rather than being limited to the explicit state-action pairs present in the dataset. This allows SCQL to exploit the structure of the state-constrained MDP more effectively and learn a better policy, with less required data. In the next section, this improvement is illustrated through an example which shows SCQL excelling even with a limited dataset.

### 3.2   An illustrative example: maze navigation

To elucidate the advantages of SCQL in comparison to BCQL, we explore a simple maze environment backed by a dataset with few trajectories. This pedagogical illustration aims to show how SCQL utilises the concept of state reachability - which enables improved performance over BCQL (which has no mechanism to utilise state reachability). Figure 2 shows the full maze environment, visualised as a 10 by 10 grid. The available states are shown as white squares, while the red squares represent the maze walls. Each state in this environment is denoted by the $(x, y)$ coordinates within the maze, and the agent's available actions comprise of simple movements: left, right, up, and down. With each movement, the agent incurs a reward penalty of a small negative value ($r_{\text{pen}} = -0.1$), but upon successfully navigating to the gold star, it is rewarded with a large positive value ($r_{\text{goal}} = 10$). Consequently, the agent's primary objective becomes devising the most direct route to the star. In this deterministic environment, the outcomes of transitioning to a state and executing an action are congruent, meaning Eq. (1) and (2) mirror each other. This allows for a direct comparison between BCQL and SCQL in terms of their performance and ability to leverage limited datasets. This experiment converges after 100 steps with a learning rate of $\alpha = 0.25$ and a discount factor of $\gamma = 0.99$.

Figure 2a provides a visual representation of the maze and accompanying dataset. Here, trajectories start at the white circle and end at the black circle. Despite its simplicity, the dataset contains a scant number of states wherein multiple actions have been taken. Within this specific framework, the notion of state reachability is straightforward: any state that is just one block away from the current state is considered reachable. Formally the state reachability is, $\mathcal{SR}_{\mathcal{M}}((x, y)) = \{(x + 1, y), (x - 1, y), (x, y + 1), (x, y - 1)\}$. Although this definition includes the wall states as reachable, SCQL can only perform updates where $s'$ is reachable and in the dataset, thus the walls are avoided. In other words, a state is considered reachable from the current state if it is one block away in any of the four cardinal directions (left, right, up, or down). This simple definition of state reachability is sufficient for demonstrating the advantages of SCQL in this illustrative example.

Applying BCQL with training as per Eq. (3) and policy extraction as per

$$\pi(s) = \max_{a \text{ s.t. } (s,a) \in \mathcal{D}} Q(s, a),$$

we obtain the policy depicted in Figure 2b, where arrows indicate policy direction. Conversely, for SCQL, training via Eq. (5) and policy training through Eq. (6) yields the policy in Figure 2c. For clarity, if

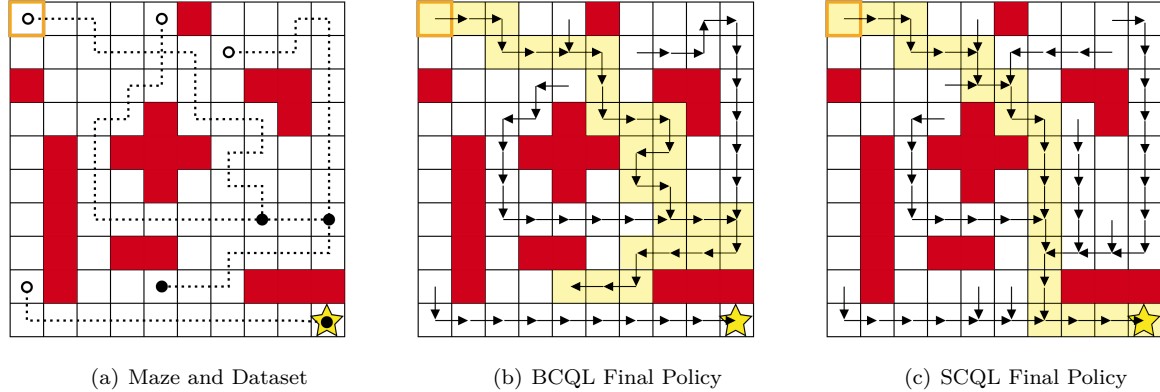

(a) Maze and Dataset         (b) BCQL Final Policy         (c) SCQL Final Policy

Figure 2: Comparison of BCQL and SCQL methods on a simple maze environment. (a) The maze is a 10 by 10 grid where the coordinate values $(x, y)$ represent the state. The high reward region is represented by a star and maze walls are represented by red grid squares. The dataset is made of 4 trajectories represented by the dotted lines where the white circle is the starting state and the black circle is the final state. (b) The final policy when applying BCQL to the dataset. (c) The final policy when applying SCQL to the dataset.

no optimal movement can be found in the state due to no information in the dataset, it is left blank. SCQL is able to leverage the sparse dataset more efficiently, enabling the agent to reach the gold star from any dataset state position. BCQL, however, succeeds in only 11 of the 57 unique dataset states, which is equivalent to the performance of behavioural cloning. SCQL's advantage stems from its ability to use state adjacency information for trajectory stitching, enabling the agent to construct optimal trajectories even in sparse datasets. By contrast, BCQL's reliance on previously observed state-action pairs limits its performance when faced with a limited dataset. In many practical settings state reachability cannot be naturally assumed from the environment. As a result, in the following sections, a deep learning implementation of SCQL is defined where state reachability is learned from the dataset.

## 4 Practical implementation: StaCQ

In this section, we aim to provide a practical implementation of SCQL. Through this method we seek to benefit from the state-constrained framework, where Q-values are updated on all state and reachable next state pairs; and the policy uses the reachable states to find a more diverse, safe and closer to optimal policy. This requires a practical method to learn state reachability, as well as an actor-critic method that exploits the benefits of the state reachability.

As a result, we introduce **StaCQ** (**Sta**te-**C**onstrained deep **Q**SS-learning), our particular deep learning implementation of **SCQL**. We present a simple, yet effective, approach to learning state reachability, using dynamics models of the environment. StaCQ is a policy constraint method that regularises the policy towards the best reachable next state in the dataset. Our specific implementation learns a QSS-function that is aligned with the underlying theory. However, state-constrained methods can be adapted from the theory in various ways, similar to how many techniques modify the batch constraint in **BCQ** (Fujimoto et al., 2019). For instance, alternative state-constrained methods might incorporate different policy regularisation techniques, such as those based on divergence measures or constraint violation penalties, or explore different approaches to estimating state reachability using available data and domain knowledge. The purpose here is to illustrate one specific implementation of the broader state-constrained framework, this implementation can be extended or modified in several ways.

### 4.1 Estimating state reachability

Central to the state-constrained approach is the concept of state reachability, as per Definition 3.1. Because the environments used in our benchmarks do not provide this information, we need to estimate reachability

from the dataset. In our implementation, we consider $\hat{s}'$ reachable to $s$ if we can predict the action that reaches $\hat{s}'$. For this, we introduce a forward dynamics model, $f_{\omega_1}(s, a)$, and an inverse dynamics model, $I_{\omega_2}(s, s')$. Both models are implemented as neural networks and trained via supervised learning on triplets $(s, a, s') \sim \mathcal{D}$. The loss function for the forward model is given by

$$\mathcal{L}_{\omega_1} = \mathbb{E}_{(s,a,s') \sim \mathcal{D}}[(f_{\omega_1}(s, a) - s')^2] \tag{7}$$

while the loss for the inverse model is:

$$\mathcal{L}_{\omega_2} = \mathbb{E}_{(s,a,s') \sim \mathcal{D}}[(I_{\omega_2}(s, s') - a)^2]. \tag{8}$$

Consistent with prior research, we adopt an ensemble strategy to account for the epistemic uncertainty (uncertainty in model parameters) (Buckman et al., 2018; Chua et al., 2018; Janner et al., 2019; Argenson & Dulac-Arnold, 2020; Yu et al., 2020; 2021). Each model within the ensemble possesses a distinct set of parameters. The ensemble's final predictions for $s'$ and $a$ are computed by taking the average of their respective outputs. It is important to note that these models are relatively simple, and incorporating more complex architectures, such as those proposed in Zhang et al. (2021), could significantly improve both model predictions and state reachability estimates.

Using these models, we propose an estimator of state reachability, denoted by $\widehat{\mathcal{SR}}_{\mathcal{M}}$, as:

$$\hat{s}' \in \widehat{\mathcal{SR}}_{\mathcal{M}}(s) \quad \text{iff} \quad ||f_{\omega_1}(s, I_{\omega_2}(s, \hat{s}')) - \hat{s}'||_\infty \leq \epsilon. \tag{9}$$

This criterion suggests that $\hat{s}'$ is reachable from $s$ only if an action can be predicted that transitions the agent from $s$ to $\hat{s}'$. Our practical implementation of state reachability, shown in Eq. (9), aims to satisfy Definition 3.1. Without access to the true environment dynamics, we must model these transitions using available data. We use the $L^\infty$-norm between observed and predicted states to ensure reachability is determined uniformly across all state dimensions. While other norms, such as $L^1$ and $L^2$, can be used, our experiments showed no significant difference in the policy produced by StaCQ between the $L^2$ and $L^\infty$ norms. An ablation study on the effects of norm choice and threshold distance is provided in Appendix F.

It is important to note that $\epsilon$, in Eq. (9), is a small positive value used to account for potential model inaccuracies, and we set $\epsilon = 0.1$ for all datasets. In Appendix B, we also present a method for reducing the computational complexity of this calculation.

## 4.2 StaCQ

StaCQ uses an actor-critic framework, where the critic, represented by the $Q(s, s')$ value, is trained by minimising the mean square error (MSE) between predicted and true QSS-values. In line with the theory, the QSS-values are updated across all pairs of states and their reachable next states:

$$\mathcal{L}_\theta = \mathbb{E}_{\substack{s \sim \mathcal{D} \\ \hat{s}' \in \widehat{\mathcal{SR}}_{\mathcal{M}}(s)}} \left[ \left( r(s, \hat{s}') + \gamma Q_{\theta'}(\hat{s}', f_{\omega_1}(\hat{s}', \pi_{\phi'}(\hat{s}'))) - Q_\theta(s, \hat{s}') \right)^2 \right]. \tag{10}$$

In this equation, the target actor produces an action, $\pi_{\phi'}(\hat{s}')$, which is passed into the forward model $f_{\omega_1}$ before being evaluated by the QSS-function. The specific reward for the reachable pair $(s, \hat{s}')$ may be unseen in the dataset, and if the reward function is unknown, it must be approximated using a neural network. Similar to the forward and inverse dynamics models, this can be achieved through supervised learning by minimising the MSE between the predicted and actual rewards:

$$\mathcal{L}_{\omega_3} = \mathbb{E}_{(s,r,s') \sim \mathcal{D}}[(r_{\omega_3}(s, s') - r)^2]. \tag{11}$$

Past research suggests that this type of reward model is effective at estimating rewards for previously unseen transitions (Hepburn & Montana, 2024). In Eq. (10), $\theta'$ represents the parameters of the target Q-value, which are incrementally updated towards $\theta$: $\theta' \leftarrow \tau\theta + (1 - \tau)\theta'$, where $\tau$ is the soft update coefficient.

To train the deterministic actor, $\pi : \mathcal{S} \to \mathcal{A}$, we aim to maximise the current QSS-values while staying close to the best next state in the dataset. Therefore, the loss we minimise is:

$$\mathcal{L}_\phi = \mathbb{E}_{s \sim \mathcal{D}} \left[ -\lambda Q_\theta(s, f_{\omega_1}(s, \pi_\phi(s))) + (f_{\omega_1}(s, \pi_\phi(s)) - \hat{s}')^2 \right], \tag{12}$$

where the hyperparameter $\lambda$ controls the level of regularisation. The $\hat{s}'$ represents the best reachable next state from $s$ according to the QSS-value,

$$\hat{s}' = \underset{s' \in \widehat{\mathcal{SR}}_\mathcal{M}(s)}{\arg\max} \ Q_\theta(s, s').$$

This is, therefore, a similar policy extraction method to TD3+BC (Fujimoto & Gu, 2021), as both methods use a MSE regulariser. However, StaCQ leverages the state reachability metric, allowing the maximisation to be performed over multiple states, whereas TD3+BC is limited to a single state-action pair.

---

**Algorithm 1** StaCQ

1: **Input:** Dataset $\mathcal{D}$, $T$ iterations, $\tau$
2: **Initialise:** $\omega_1$, $\omega_2$, $\omega_3$, $\theta$, $\theta'$, $\phi$
3: Pre-train $f_{\omega_1}$ & $I_{\omega_2}$: Eqs.(7) & (8)
4: Pre-train reachability criteria $\widehat{\mathcal{SR}}_\mathcal{M}$
5: **for** $t = 1, \ldots, T$ **do**
6:     Optimise reward function: Eq. (11)
7:     Optimise QSS-value: Eq. (10)
8:     Optimise policy: Eq. (12)
9:     Update target networks:
      $\theta' \leftarrow \tau\theta + (1 - \tau)\theta'$,
      $\phi' \leftarrow \tau\phi + (1 - \tau)\phi'$
10: **end for**

---

Similar to batch-constrained offline RL methods, StaCQ aims to balance staying close to the dataset while maximising the Q-value. The hyperparameter $\lambda$ controls the trade-off between producing actions that lead to high-value OOD states and staying close to states in the dataset, specifically the highest value reachable state. Since the forward model is fixed, a small amount of Gaussian noise is added to the policy action, to ensure the policy does not overfit to the forward model, resulting in a more robust policy. The full algorithm is presented in Algorithm 1. Notably, all three constituent models are trained using supervised learning, making them straightforward sub-processes within the overall architecture. The code can be found at `https://github.com/CharlesHepburn1/State-Constrained-Offline-Reinforcement-Learning`.

## 5 Related work

**Model-free offline RL.** BCQ has emerged as one of the pioneering offline deep RL methodologies (Fujimoto et al., 2019). It posits that having all $(s, a)$ pairs in the MDP enables the construction of an optimal QSA-value by focusing updates only on the $(s, a)$ pairs present in the dataset. As a result, BCQ restricts the policy to these observed $(s, a)$ pairs while allowing minimal perturbations to facilitate marginal improvements. Following BCQ, numerous algorithms have been introduced to develop alternative ways to constrain the policy to seen $(s, a)$ pairs, either by restricting it to the support of the $(s, a)$-distribution (Kumar et al., 2019; Wu et al., 2019; Siegel et al., 2020; Kostrikov et al., 2021a; Zhou et al., 2021; Brandfonbrener et al., 2021) or directly limiting it to the $(s, a)$ pairs (Fujimoto & Gu, 2021).

In contrast to policy constraints, some methods tackle distributional shifts by pessimistically evaluating the value function on OOD $(s, a)$ pairs (Kumar et al., 2020; Yu et al., 2021; An et al., 2021). There is also a subset of offline RL methods that combines both pessimistic value evaluation and policy constraints (Dadashi et al., 2021; Beeson & Montana, 2024). Alternatively, implicit Q-learning (IQL) (Kostrikov et al., 2021b) and policy-guided offline RL (POR) (Xu et al., 2022) aim to learn an optimal value function through expectile regression, with POR estimating a state-value function and IQL estimating an action-value function. Both methods learn the optimal policy (referred to as the guide policy in POR) via advantage-weighted regression. All of these approaches adopt the batch-constrained method, prioritising explicit $(s, a)$ pairs or $(s, s')$ pairs found in the dataset over unobserved actions. Our state-constrained method, however, relaxes the batch-constrained objective by requiring constraints only on reachable states in the dataset rather than explicit $(s, a)$ pairs.

**Model-based offline RL.** Model-based methods learn the dynamics of the environment to support policy learning (Sutton, 1991; Janner et al., 2019). There are generally two main approaches to model-based

offline RL. The first approach learns a pessimistic model of the environment and performs rollouts with this augmented model, effectively increasing the dataset size (Yu et al., 2020; Kidambi et al., 2020; Rigter et al., 2022). The second approach uses the models for planning, allowing the agent to look ahead during evaluation to determine the optimal path (Argenson & Dulac-Arnold, 2020; Zhan et al., 2021; Janner et al., 2022; Diehl et al., 2021). In contrast, StaCQ uses models to implement our concept of state reachability; thus, StaCQ does not generate new states or involve planning during evaluation.

**State reachability.** The state-constrained approach relies heavily on the concept of state reachability. Prior studies on state reachability (Hepburn & Montana, 2022; 2024) have employed a Gaussian distribution to determine the likelihood of stitching from $s$ to $s'$. In contrast, our method treats a state as either reachable or not, avoiding the use of continuous probability distributions. State reachability is closely related to state similarity, a well-explored concept in the literature (Zhang et al., 2020; Agarwal et al., 2021; Le Lan et al., 2021). One way to measure state similarity is through bisimulation metrics, which are based on the environment's dynamics (Ferns et al., 2012). Traditional bisimulation methods often require full state enumeration (Chen et al., 2012; Bacci et al., 2013a;b), leading to the development of more scalable pseudometric approaches (Castro, 2020). While incorporating a pseudometric into our state reachability framework could be a potential avenue, we leave this for future work. Our current state reachability metric is simple, intuitive, and grounded in the core definition.

**Trajectory stitching.** A key feature of offline RL is its ability to stitch together trajectories (Kostrikov et al., 2021b), i.e., combining previously observed trajectories to create a novel one that completes the task. Imitation learning methods, such as behavioural cloning (Pomerleau, 1988; 1991), struggle with stitching due to their inability to distinguish between optimal and sub-optimal states. Similarly, the Decision Transformer (DT) (Chen et al., 2021) formulates the offline RL problem as a supervised learning task using a goal-conditioned policy. While DT exhibits weak stitching capabilities, efforts have been made to address this by integrating offline RL principles (Yamagata et al., 2023; Wu et al., 2023). Combining DT with StaCQ's policy extraction approach could yield further improvements in stitching. However, our current work focuses on a straightforward implementation of state-constrained offline RL.

## 6 Experimental results

We evaluate StaCQ against several model-free and model-based baselines on the D4RL benchmarking datasets from the OpenAI Mujoco tasks (Todorov et al., 2012; Fu et al., 2020). The model-free baselines we compare against are BCQ (Fujimoto et al., 2019), TD3+BC (Fujimoto & Gu, 2021), and IQL (Kostrikov et al., 2021b). BCQ is the foundational batch-constrained offline RL method on which many subsequent methods are based, making it the most direct theoretical comparison. TD3+BC is the most similar to StaCQ in terms of implementation, as both use a BC-style regularisation process. IQL, a current SOTA model-free method, provides a strong comparison point.

The model-based baselines include MBTS (Hepburn & Montana, 2024), Diffuser (Janner et al., 2022), and RAMBO (Rigter et al., 2022). MBTS employs a data augmentation strategy with a different state reachability approach compared to StaCQ. Diffuser is a planning-based method that uses a diffusion model, making its dynamics model more complex than StaCQ's. RAMBO, a SOTA model-based method, performs rollouts using a pessimistic dynamics model of the environment. Although StaCQ does not use models for planning or rollouts, we include these model-based algorithms for comparison.

Additionally, we devised a one-step version of StaCQ (Brandfonbrener et al., 2021), detailed in Appendix D, to showcase the flexibility of the state-constrained framework.

Results for both StaCQ and the one-step version of StaCQ are shown in Table 1. For the Antmaze tasks, it is unnecessary to train a reward model since these tasks feature sparse rewards, meaning the reward for being in $s'$ remains constant. As all $s'$ are in the dataset, the reward, $r(s, s')$, is always known for all $s$ where $s' \in \mathcal{SR}_{\mathcal{M}}(s)$. Also for the Antmaze tasks, StaCQ deploys independent target Q-value estimates. This is a techniques from Ghasemipour et al. (2022) that improves the performance of the policy in the medium and large Antmaze tasks. Despite potential model errors in estimating state reachability, StaCQ performs

Table 1: Average normalised scores on the D4RL datasets. StaCQ results have been obtained by taking an average over 5 seeds. The bolded scores are within 95% of the highest performing method.

| | | Model-free baselines | | | | Model-based baselines | | | Our methods | |
|---|---|---|---|---|---|---|---|---|---|---|
| | | BC | BCQ | TD3+BC | IQL | MBTS | Diffuser | RAMBO | StaCQ | OneStep StaCQ |
| Hopper | Rand | 6.2 | 7.6 | 8.5 | - | - | - | **21.6** | $17.5 \pm 12.8$ | $7.5 \pm 0.4$ |
| | Med-Rep | 22.5 | 51.0 | 60.9 | **94.7** | 50.2 | 93.6 | **96.6** | $\mathbf{99.1 \pm 1.3}$ | $\mathbf{99.4 \pm 0.6}$ |
| | Med | 56.8 | 60.9 | 59.3 | 66.3 | 64.3 | 74.3 | 92.8 | $\mathbf{100.2 \pm 3.0}$ | $93.3 \pm 3.2$ |
| | Med-Exp | 54.2 | 85.9 | 98.0 | 91.5 | 94.8 | 103.3 | 83.3 | $\mathbf{111.9 \pm 0.2}$ | $92.1 \pm 9.2$ |
| Walker2D | Rand | 1.4 | 4.4 | 1.6 | - | - | - | **11.5** | $4.4 \pm 6.4$ | $6.5 \pm 0.9$ |
| | Med-Rep | 25.5 | 60.7 | 81.8 | 73.9 | 61.5 | 70.6 | **85.0** | $\mathbf{87.2 \pm 7.3}$ | $\mathbf{88.1 \pm 5.3}$ |
| | Med | 39.4 | 73.7 | 83.7 | 78.3 | 78.8 | 79.6 | 86.9 | $\mathbf{92.2 \pm 3.0}$ | $85.7 \pm 10.4$ |
| | Med-Exp | 90.5 | 94.5 | **110.1** | **109.6** | 108.8 | 106.9 | 68.3 | $\mathbf{116.2 \pm 1.5}$ | $107.6 \pm 8.5$ |
| Halfcheetah | Rand | 2.1 | 2.2 | 11.0 | - | - | - | **40.0** | $24.3 \pm 1.3$ | $2.6 \pm 1.6$ |
| | Med-Rep | 34.5 | 41.1 | 44.6 | 44.2 | 39.8 | 37.7 | **68.9** | $52.2 \pm 0.5$ | $46.4 \pm 0.4$ |
| | Med | 42.4 | 46.6 | 48.3 | 47.4 | 43.2 | 42.8 | **77.6** | $57.6 \pm 0.6$ | $50.0 \pm 0.2$ |
| | Med-Exp | 66.6 | 87.8 | 90.7 | 86.7 | 86.9 | 88.9 | **93.7** | $\mathbf{96.4 \pm 2.9}$ | $\mathbf{94.9 \pm 1.0}$ |
| Total (excl. rand) | | 432.4 | 602.2 | 677.4 | 692.6 | 628.3 | 697.7 | 753.1 | **813.0** | 757.5 |
| Antmaze | Umaze | 53.4 | 70.0 | 78.6 | **87.5** | - | - | 25.0 | $\mathbf{89.8 \pm 5.0}$ | $75.6 \pm 3.8$ |
| | U-diverse | 64.6 | 44.0 | **71.4** | 62.2 | - | - | 0.0 | $64.0 \pm 29.7$ | $67.2 \pm 13.8$ |
| | M-play | 0.0 | 0.0 | 3.0 | **71.2** | - | - | 16.4 | $47.2 \pm 27.1$ | $20.6 \pm 12.5$ |
| | M-diverse | 0.8 | 0.0 | 10.6 | **70.0** | - | - | 23.2 | $47.4 \pm 27.6$ | $13.0 \pm 4.2$ |
| | L-play | 0.0 | 0.0 | 0.0 | **39.6** | - | - | 0.0 | $31.4 \pm 6.0$ | $5.2 \pm 5.2$ |
| | L-diverse | 0.0 | 0.0 | 0.2 | **47.5** | - | - | 2.4 | $40.0 \pm 22.4$ | $2.2 \pm 1.6$ |
| Total (Antmaze) | | 118.8 | 114.0 | 163.8 | **378.0** | - | - | 67.0 | 319.8 | 183.8 |

remarkably well against the baselines. The learning curves for all tasks in Table 1 are shown in Appendix G.

StaCQ outperforms all model-free baselines in the locomotion tasks (Hopper, Halfcheetah, and Walker2d) across the entire range of datasets (random, medium-replay, medium, and medium-expert). These environments are complex robotics tasks, with the random and medium-replay consisting of few good-quality episodes. StaCQ also surpasses most model-based baselines in the locomotion tasks, with the exception of RAMBO, which outperforms StaCQ in the Halfcheetah and random tasks. However, on average StaCQ is the highest-performing method across all locomotion tasks, consistently ranking as either the top or second-best performing method. TD3+BC uses a consistent hyperparameter across all datasets, whereas our StaCQ methods use an environment-dependent hyperparameter (see Appendix C). To enable a fair comparison, Appendix E compares TD3+BC and StaCQ using the same hyperparameter tuning strategy.

Through utilising the higher number of reachable states and thus finding higher quality OOD actions that lead to in-distribution states, StaCQ performs well across most of the locomotion tasks. However its performance in the Antmaze tasks is comparatively lower. StaCQ does outperform RAMBO in the Antmaze tasks, which are notoriously challenging for methods involving dynamics models (Wang et al., 2021; Rigter et al., 2022), and performs competitively against the model-free baselines. Notably, StaCQ outperforms BCQ and TD3+BC, which are the most comparable baselines in terms of both theory and implementation. Methods such as IQL, which rely on advantage-weighted policy extraction, may have an advantage in these tasks due to their ability to handle sparse rewards more effectively. Nonetheless, StaCQ is either the highest or second highest performing method on the Antmaze tasks against the comparative methods.

## 7 Discussion and conclusions

In this paper, we introduced a method for state-constrained offline RL. Most prior offline RL approaches are batch-constrained, limiting the learning of the Q-function or policy to $(s, a)$ pairs or the $(s, a)$-distribution in the dataset. In contrast, our state-constrained methodology confines learning updates to states within the dataset, offering significant potential reductions in dataset size required for achieving optimality. By focusing on states rather than state-action pairs, the state-constrained approach enables more efficient learning updates and can achieve optimality with smaller datasets compared to batch-constrained methods.

Central to the proposed state-constrained approach is the concept of state reachability, which we define and illustrate in a simple maze environment. This example underscores the improved stitching capability of state-constrained methods over batch-constrained ones, particularly in datasets with limited trajectories. Additionally, we provide theoretical backing that affirms the potential for our method's convergence based on dataset states rather than state-action pairs. Similar to BCQ (Fujimoto et al., 2019), our theory assumes a deterministic MDP. This focus on deterministic environments simplifies the theoretical development (Fujimoto et al., 2019; Jin et al., 2021; Shi et al., 2022), and provides a fair comparison with the BCQ theory, while also providing clear guarantees for convergence, making it an appropriate starting point for the state-constrained approach. Our theoretical goal is to demonstrate convergence and provide a comparison with BCQ. As a result, Theorem 3.6 shows that SCQL converges to the optimal policy with a smaller requisite dataset size than BCQ; and, Theorem 3.7 shows that the policy produced by SCQL can always be preferred to the policy produced by BCQL.

Based from these theoretical developments, our practical implementation also defines reachable states deterministically — states are considered reachable or not, regardless of potential stochastic transitions. Deterministic settings align with many practical applications, such as robotic control or industrial systems, where transitions can be accurately modelled. However, some real-world environments are stochastic, where actions lead to a distribution of possible next states. Future work will extend state reachability to account for stochastic transitions by incorporating probabilistic models of the environment's dynamics. Theoretically, the definition of state reachability can be extended to stochastic MDPs, for example $s'$ is reachable from $s$ if it is the most likely next state when performing action $a$, i.e $s' \in \mathcal{SR}_{\mathcal{M}}(s)$ if $p(s'|s, a) = \max_{s'_i \sim p(\cdot|s, a)} p(s'_i|s, a)$. Practically, this will involve redefining reachability in terms of the likelihood of reaching a state under certain actions, possibly using techniques such as probabilistic graphical models. The key challenge will be maintaining computational efficiency while handling the increased complexity of probabilistic transitions. Extending the framework in this way will allow the state-constrained approach to be applied to more complex, uncertain environments, improving its robustness and real-world applicability.

We also introduced *StaCQ*, an implementation of state-constrained deep offline RL. StaCQ leverages QSS-learning (Edwards et al., 2020) to estimate the value of transitioning from $s$ to $s'$, which is particularly advantageous in a state-constrained context as it avoids action dependency. StaCQ determines state reachability using forward and inverse dynamics models, trained via supervised learning on the dataset. While the performance of these models can influence StaCQ, the algorithm consistently demonstrates strong performance in complex environments, often surpassing baseline batch-constrained methods. StaCQ relies on an ensemble of forward and inverse dynamics models. Increasing the size of the ensemble would enable more accurate estimates, however the number of models is consistent with prior research (Janner et al., 2019) and is accurate enough for StaCQ to perform well, as shown in Table 1. StaCQ uses an ensemble of four critics whereas OneStep StacQ uses a single critic. These values were chosen as they enable stable estimates across all datasets.

There are several potential improvements to explore in future work. Despite the simplicity of the models we used, our reachability measure delivered SOTA performance. Future enhancements could improve this by adding more complexity to the dynamics models, reducing model error, and capturing the true environment dynamics more accurately. Alternatively, developing entirely new state reachability criteria independent of dynamics models could significantly enhance the algorithm's effectiveness. Potential ideas include leveraging graph-based approaches or using reinforcement learning techniques to learn reachability directly from data. Additionally, the policy extraction process could be improved by incorporating elements such as a Decision Transformer (Chen et al., 2021) or diffusion policies (Ajay et al., 2022). These techniques have shown

promise in offline RL by leveraging the structure of the dataset to generate more diverse and effective policies. Integrating them into StaCQ could allow the algorithm to exploit the available data more effectively.

Similar to batch-constrained methods, further advances might arise from relaxed state constraints, albeit with tailored considerations for this setting. Using ensembles to assess the uncertainty of QSA-values (An et al., 2021; Ghasemipour et al., 2022; Beeson & Montana, 2024) and constraining actions to specific regions has proven advantageous in the batch-constrained domain. Similar methodologies could be extended to state-constrained approaches, such as employing ensembles of QSS-functions to evaluate uncertainty.

Beyond improving state reachability estimates and policy extraction mechanisms, the potential for advancement in state-constrained methodologies is vast. One intriguing direction is to refine state reachability estimates to focus solely on states, excluding action predictions. Such advancements could lead to algorithms that rely entirely on state-only datasets, which is particularly relevant for real-world scenarios like learning from video data, where states are observable (as images), but the actions leading to transitions are unknown. Expanding the notion of state reachability is another promising avenue. Our current definition, based on one-step state reachability ($\exists a$ such that $p(s'|s, a) = 1$), could be extended to multiple steps. For example, a two-step reachability could be defined as $s'' \in \mathcal{SR}_{\mathcal{M}}(s)$ if $\exists a_1, a_2$ such that $p(s'|s, a_1) = 1$ and $p(s''|s', a_2) = 1$. This principle can be further extrapolated, providing a foundation for novel model-based offline RL algorithms. Extending this notion of state reachability will allow for a higher number of reachable states to be found, while being sure they lead, in $k$-steps, to in-distribution states. Appendix H shows the theoretical benefits of a multi-step state reachable method on a simple maze environment. However, a practical method that incorporates model error is left for future work. Lastly, the state-constrained methodology holds significant potential for multi-agent offline RL scenarios, where complexity grows with the action space size. By reducing action dependence, the state-constrained approach appears well-suited to tackle such challenges.

In conclusion, we have ventured into the realm of state-constrained offline RL, introducing a novel paradigm that emphasises states over state-action pairs in the dataset. While this paper lays the foundational groundwork, the path forward calls for further investigation, refinement, and integration with other emerging techniques. We believe that these endeavours can push the boundaries of current offline RL methodologies and offer unprecedented solutions to complex problems.

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

## A  Missing proofs

**Proof of Theorem 3.2**  Since each $(s, s')$ pair is visited infinitely often, consider consecutive intervals during which each $(s, s')$ transition occurs at least once. We want to show the max error over all entries in the Q table is reduced by at least a factor of $\gamma$ during each such interval.

Let $\Delta_n$ by the max error in $Q_n$, $\Delta_n := \max_{s,s'} |Q_n(s, s') - Q^*(s, s')|$. Then,

$$|Q_{n+1}(s, s') - Q^*(s, s')| = |(r(s, s') + \gamma \max_{s''} Q_n(s', s'') - (r(s, s') + \gamma \max_{s''} Q^*(s', s'')))| \tag{13}$$

$$= \gamma |\max_{s''} Q_n(s', s'') - \max_{s''} Q^*(s', s'')| \tag{14}$$

$$= \gamma |\max_{s''} Q_n(s', s'') - \max_{s''} Q^*(s', s'')| \tag{15}$$

$$\leq \gamma \max_{s''} |Q_n(s', s'') - Q^*(s', s'')| \tag{16}$$

$$\leq \gamma \max_{\hat{s},s''} |Q_n(\hat{s}, s'') - Q^*(\hat{s}, s'')| \tag{17}$$

Which implies $|Q_{n+1}(s, s') - Q^*(s, s')| \leq \gamma \Delta_n$.  □

Where for brevity the maximisation $\max_{s''}$ is the shorthand for $\max_{s'' \text{ s.t. } s'' \in \mathcal{SR}_\mathcal{M}(s')}$. Also, Eq. (13) is using the training rule; Eqs. (14) and (15) are simplifying and rearranging; (16) uses the fact that $|\max_x f_1(x) - \max_x f_2(x)| \leq \max_x |f_1(x) - f_2(x)|$; Eq. (17) we introduced a new variable $\hat{s}$ for which the maximisation is performed - this is permissible as allowing this additional variable to differ will always be at least the maximum value.

Thus, the updated $Q_{n+1}(s, s')$ for any $s, s'$ is at most $\gamma$ times the maximum error in the $Q_n$ table, $\Delta_n$. The largest error in the initial table, $\Delta_0$, is bounded because the values of $Q_0(s, s')$ and $Q^*(s, s')$ are bounded $\forall s, s'$. Now, after the first interval during which each $s, s'$ is visited the largest error will be at most $\gamma \Delta_0$. After $k$ such intervals, the error will be at most $\gamma^k \Delta_0$. Since each state is visited infinitely often, the number of such intervals is infinite and $\Delta_n \to 0$ as $n \to \infty$.

**Proof of Theorem 3.4.**  1.Define deterministic state-constrained MDP $\mathcal{M}_\mathcal{S}$. We define $\mathcal{M}_\mathcal{S} = (\mathcal{S}, \mathcal{A}, \mathcal{P}_\mathcal{S}, \mathcal{R}, \gamma)$ where $\mathcal{S}$ and $\mathcal{A}$ are the same as the original MDP. We have the transition probability where

$$p_\mathcal{S}(s'|s, a) = \begin{cases} 1 & \text{if } (s, s' \in \mathcal{D} \text{ and } s' \in \mathcal{SR}_\mathcal{M}(s)) \text{ or } (s \notin \mathcal{D} \text{ and } s' = s_{\text{terminal}}) \\ 0 & \text{otherwise.} \end{cases}$$

this is a deterministic transition probability for $s, s'$ that are in the dataset and reachable. If a pair exists but is not in the dataset we set the rewards to be the initialised $Q(s, s')$ values, otherwise they have the reward seen in the dataset.

2. We have all the same assumptions under $\mathcal{M}_\mathcal{S}$ as we do under $\mathcal{M}$, apart from infinite $(s, s')$ visitation, so we need that and then it follows from Theorem 3.2.

Note that sampling under the dataset $\mathcal{D}$ with uniform probability satisfies the infinite state-next-state visitation assumptions of the MDP $\mathcal{M}_\mathcal{S}$. For a reachable pair $(s, s') \notin \mathcal{D}$ [this may be due to $s \notin \mathcal{D}$ or $s' \notin \mathcal{D}$], $Q(s, s')$ will never be updated and will correspond to the initialised value. So sampling from $\mathcal{D}$ is equivalent to sampling from the MDP $\mathcal{M}_\mathcal{S}$ and QSS-learning converges to the optimal value under $\mathcal{M}_\mathcal{S}$ by following Theorem 3.2.  □

**Proof of Theorem 3.5.**  This follows from Theorem 3.2, noting the state-constraint is non-restrictive with a dataset which contains all possible states.  □

**Proof of Theorem 3.6.**  This result follows from the fact that if every $s$ is visited infinitely then due to the definition of state reachability we can evaluate over all $(s, s')$-pairs in the dataset. Then following Theorem 3.4, which states QSS-learning learns the optimal value for the MDP $\mathcal{M}_\mathcal{S}$ for $s, s' \in \mathcal{D}$. However, the deterministic $\mathcal{M}_\mathcal{S}$ corresponds to the original $\mathcal{M}$ in all seen state and reachable next-state pairs. Noting

that state-constrained policies operate only on $s, s' \in \mathcal{D}$, where $\mathcal{M}_{\mathcal{S}}$ corresponds to the true MDP, it follows that $\pi^*$ will be the optimal state-constrained policy from the optimality of QSS-learning. $\qquad\square$

**Theorem A.1.** *In a deterministic setting, QSA-values are equivalent to QSS-values*

*Proof.* See Theorem 2.2.1 in (Edwards et al., 2020) $\qquad\square$

**Proof of Theorem 3.7.** Case 1: In this situation, for all states in the dataset, we do not have any extra reachable states, other than the pairs i.e. $\forall s \in \mathcal{D}, \mathcal{SR}_{\mathcal{M}}(s) = \{s'\}$, where $(s, s') \in \mathcal{D}$. In this case the state-constrained MDP, Definition 3.3, is equivalent to the batch-constrained MDP in (Fujimoto et al., 2019). The condition of the state-constrained MDP,

$$s, s' \in \mathcal{D} \text{ and } s' \in \mathcal{SR}_{\mathcal{M}}(s)$$

becomes $(s, s') \in \mathcal{D}$ as $s' \in \mathcal{SR}_{\mathcal{M}}(s)$ only exists where $(s, s') \in \mathcal{D}$. From this, the probability transition function for the state-constrained and batch-constrained MDPs are equivalent and thus so are the MDPs themselves. Finally, for this case we just need to show that the SCQL and BCQL updates are the same. So under this case condition the SCQL update becomes

$$Q(s, s') \leftarrow (1 - \alpha)Q(s, s') + \alpha\left[r(s, s') + \gamma \max_{s'' \text{ s.t. } (s', s'') \in \mathcal{D}} Q(s', s'')\right].$$

From Theorem A.1, for a transition $(s, a, s') \in \mathcal{D}$ we have $Q(s, a) = Q(s, s')$ and thus for the transition $(s', a', s'') \in \mathcal{D}$ we have $Q(s', a') = Q(s', s'')$, therefore the SCQL update is equivalent to

$$Q(s, a) \leftarrow (1 - \alpha)Q(s, a) + \alpha\left[r(s, a) + \gamma \max_{a' \text{ s.t. } (s', a') \in \mathcal{D}} Q(s', a')\right].$$

Therefore, for case 1 BCQL and SCQL have the same Q-value update and therefore have the same optimal policy. So, $\pi_{\text{BCQL}}(s) = \pi_{\text{SCQL}}(s), Q(s, \pi_{\text{BCQL}}(s)) = Q(s, \pi_{\text{SCQL}}(s))$, and thus $V_{\pi_{\text{BCQL}}}(s) = V_{\pi_{\text{SCQL}}}(s), \forall s \in \mathcal{D}$.

Case 2: In this case, for a single state in the dataset, we have one reachable next state that is unseen as a pair in the dataset, i.e. $\exists \tilde{s}, \hat{s}' \in \mathcal{D}$ s.t. $\hat{s}' \in \mathcal{SR}_{\mathcal{M}}(\tilde{s})$ and $(\tilde{s}, \hat{s}') \notin \mathcal{D}$. In this case the state-constrained MDP probability transition function becomes: for $a = I(s, s') \in \mathcal{A}$

$$p_{\mathcal{S}}(s'|s, a) = \begin{cases} 1 & \text{if } ((s, s') \in \mathcal{D}) \text{ or } (s \notin \mathcal{D} \text{ and } s' = s_{\text{terminal}}) \text{ or } (s = \tilde{s} \text{ and } s' = \hat{s}') \\ 0 & \text{otherwise.} \end{cases}$$

This is the transition function for the batch-constrained MDP but allowing for an extra transition from $\tilde{s}$ to $\hat{s}'$. Now comparing BCQL and SCQL Q-value updates, we have all equivalent values for the transition $(s, a, s')$ except for the trajectory that contains the state $\tilde{s}$. For the converged QSS-value, $Q^*$, let $\tilde{s}_{-1}$ be the state in the trajectory previous to $\tilde{s}$ and $\tilde{s}'$ be the next state after $\tilde{s}$ in the trajectory,

$$Q^*(\tilde{s}_{-1}, \tilde{s}) \leftarrow (1 - \alpha)Q^*(\tilde{s}_{-1}, \tilde{s}) + \alpha\left[r(\tilde{s}_{-1}, \tilde{s}) + \gamma \max\{Q^*(\tilde{s}, \tilde{s}'), Q^*(\tilde{s}, \hat{s}')\}\right]$$

If $Q^*(\tilde{s}, \tilde{s}') \geq Q^*(\tilde{s}, \hat{s}')$ then again BCQL is equivalent to SCQL. However, if $Q^*(\tilde{s}, \tilde{s}') < Q^*(\tilde{s}, \hat{s}')$ then SCQL will have higher values than BCQL for all states previous to $\tilde{s}$ and therefore will produce a higher quality policy by the policy improvement theorem. In this case, as $Q^*(\tilde{s}, \tilde{s}') < Q^*(\tilde{s}, \hat{s}')$, $\exists s$ such that $Q(s, \pi_{\text{SCQL}}(s)) > Q(s, \pi_{\text{BCQL}}(s))$, thus $V_{\pi_{\text{SCQL}}}(s) \geq V_{\pi_{\text{BCQL}}}(s), \forall s \in \mathcal{D}$.

Then without loss of generality Case 2 can be extended to all cases where we have multiple reachable next states for multiple dataset states, that have higher value according to the optimal QSS-value. $\square$

## B   Reducing the complexity of state reachability estimation

Directly computing $\widehat{\mathcal{SR}_{\mathcal{M}}}$ for every state $s$ would entail comparing each state in the dataset against all others, an approach that is computationally prohibitive for larger datasets. To address this challenge, we calculate the range of potentially reachable states for each state dimension. This calculation is based on a set of random actions, denoted as $\{a_{\text{rand}}^i\}_{i=1}^n$.

For a given state $s$, we first determine its range by calculating the minimum and maximum values using $f_{\omega_1}$. Specifically, the minimum range $R_{\min}(s)$ is computed as $\min_i f_{\omega_1}(s, a_{\text{rand}}^i)$, and the maximum range $R_{\max}(s)$ is $\max_i f_{\omega_1}(s, a_{\text{rand}}^i)$. Subsequently, we construct a smaller set of states within the range $(R_{\min}(s), R_{\max}(s))$, using an R-tree (Guttman, 1984), a data structure that efficiently identifies states within a specified hyper-rectangle (our range). This approach, leveraging the R-tree's efficient search capability, dramatically reduces the size of the dataset. Consequently, the models are applied only to this refined set of states, leading to a significant reduction in computational complexity.

## C   Implementation details

For our experiments we use $\epsilon = 0.1$ for the state reachability criteria where the LHS is also scaled by the maximum of the difference in state range, i.e our state reachability estimate for state $s$ is determined by

$$\left\| \frac{f_{\omega_1}(s, I_{\omega_2}(s, s')) - s'}{R_{\max}(s) - R_{\min}(s)} \right\|_\infty < \epsilon.$$

This means that the maximum state dimension model prediction error must be within 10% of the true state range. Due to this we also normalise all states which gives more accurate model prediction and is done in the same way as TD3+BC (Fujimoto & Gu, 2021), i.e let $s_i$ be the $i$th feature of state $s$

$$s_i = \frac{s_i - \mu_i}{\sigma_i + \epsilon_s},$$

where $\mu_i$ and $\sigma_i$ are the mean and standard deviation of the $i$th feature across all states and $\epsilon_s = 10^{-3}$ is a small normalisation constant. Similar to TD3+BC, we do not normalise states for the Antmaze tasks as this is harmful for the results.

For the MuJoCo locomotion tasks we evaluate our method over 5 seeds each with 10 evaluation trajectories; whereas for the Antmaze tasks we also evaluate over 5 seeds but with 100 evaluation trajectories. Just like TD3+BC (Fujimoto & Gu, 2021), we scale our hyperparameter by the average Q-value across the minibatch

$$\lambda = \frac{\alpha}{\frac{1}{N} \sum_i Q(s_i, s_i')},$$

where $N$ is the size of the minibatch (in our case $N = 256$) and $s_i'$ is the next state where the policy action leads from $s$. We aim to find a consistent $\alpha$ hyperparameter across the different environments. We find that the optimal consistent hyperparameters are $\alpha = \{1, 5, 10\}$ for Hopper, Walker2d and Halfcheetah respectively. However for Halfcheetah -medium expert we use $\alpha = 0.5$ due to the significant improvement. For the Antmaze tasks, each maze size is a new environment and we use $\alpha = \{2, 10, 19\}$ for the umaze, medium and large environments respectively. Across all environments, we add a small amount of zero mean Gaussian noise to the policy action before being input into the forward model, we use a fixed variance of 0.1. It should be noted that making this policy noise vary across the different environments could improve results greatly, however we aim to keep this fixed so that we have a general method that can be easily optimised on other environments.

The actor, critic and reward model are represented as neural networks with two hidden layers of size 256 and ReLU activation. They are trained using the ADAM optimiser (Kingma & Ba, 2014) and have learning rates $3e - 4$, the actor also has a cosine scheduler. We use an ensemble of 4 critic networks and take the minimum value across the networks. Also we use soft parameter updates for the target critic network with parameter $\tau = 0.005$, and we use a discount factor of $\gamma = 0.99$. For the locomotion tasks we use a shared

target value to update the critic towards, whereas for the Antmaze tasks we use independent target values for each critic value. Both the inverse and forward dynamics models are represented as neural networks with three hidden layers of size 256 and ReLU activation. They are trained using the ADAM optimiser with a learning rate of $4e-3$ and a batch size of 256. We use an ensemble of 7 forward models and 3 inverse models and then take a final prediction as an average across the ensemble. Also our forward model predicts the state difference rather than the next state directly, which improves model prediction performance.

To attain the results for BCQ, we re-implemented the algorithm from the paper (Fujimoto et al., 2019) and trained on the version 2 datasets. Following the same practises as StaCQ we evaluate BCQ over 5 seeds each with 10 evaluations on the MuJoCo locomotion tasks and 5 seeds each with 100 evaluations on the Antmaze tasks. All other results in Table 1 were obtained from the original authors' papers. Our experiments were performed with a single GeForce GTX 3090 GPU and an Intel Core i9-11900K CPU at 3.50GHz. Each run takes on average 2.5 hours, where models are pre-trained and state reachability is given (the same models and reachability lists are given to each run). We provide 5 runs for each dataset (18) which gives a total run time of 225 hours.

## D    One-step method

StaCQ, Algorithm 1, that we have introduced in this paper is an actor-critic method that learns the QSS-value and policy together. Alternatively, the QSS-value can be learned directly from the data, in an on-policy fashion, then a policy can be extracted directly from this on-policy QSS-value. These approaches are known as one-step methods (Brandfonbrener et al., 2021). In this section we adapt StaCQ into a one-step method.

### D.1    Estimating QSS-values

So that the QSS-values are learned on-policy while still taking advantage of the state-constrained framework we introduce a small modification to Eq. (5). Since the pair $(s', s'')$ is unseen in the dataset, we use the following approximation:

$$\max_{\substack{s'' s.t. (s'', s''') \in \mathcal{D} \\ s'' \in \mathcal{SR}_{\mathcal{M}}(s')}} \{r(s', s'') + \gamma Q(s'', s''')\} \tag{18}$$

to replace $\max_{\substack{s'' s.t. s'' \in \mathcal{D} \\ s'' \in \mathcal{SR}_{\mathcal{M}}(s')}} Q(s', s'')$. This adjustment ensures that the QSS-value is only evaluated on explicit state-next-state pairs, thereby avoiding OOD $(s, s')$-pairs. Although this approach diverges slightly from the theoretical method, where QSS-value updates are performed on every pair, it provides a practical solution.

Using Eq. (18), the approximation of state reachability and the reward model Eq. (11), the on-policy state-constrained QSS-values can be refined by reducing the MSE between the target and the actual QSS-values. The target is determined by identifying the maximum value across reachable states:

$$\mathcal{L}_\theta = \mathbb{E}_{(s,s') \sim \mathcal{D}} \left[ \left( r(s, s') + \gamma \max_{\substack{s'' s.t. (s'', s''') \in \mathcal{D}, \\ s'' \in \mathcal{SR}_{\mathcal{M}}(s')}} \{r_{\omega_3}(s', s'') + \gamma Q_{\theta'}(s'', s''')\} - Q_\theta(s, s') \right)^2 \right]. \tag{19}$$

Here, $\theta'$ represents the parameters for a target Q-value which are incrementally updated towards $\theta$: $\theta' \leftarrow \tau\theta + (1 - \tau)\theta'$, with $\tau$ being the soft update coefficient.

### D.2    Policy extraction step

Eq. (19) gives a one-step optimal state-constrained QSS-value. However, this equation alone does not produce an optimal action. To determine the optimal action, we need to add a policy extraction step. We use the same policy extraction method as StaCQ, a state behaviour cloning regularised policy update similar to TD3+BC (Fujimoto & Gu, 2021):

$$\mathcal{L}_\phi = \mathbb{E}_{s \sim \mathcal{D}} \left[ \lambda Q_\theta(s, f_{\omega_1}(s, \pi_\phi(s))) + (f_{\omega_1}(s, \pi_\phi(s))) - \hat{s}')^2 \right], \tag{20}$$

where

$$\hat{s}' = \underset{\substack{s' \text{ s.t.} (s',s'') \in \mathcal{D} \\ s' \in \widehat{\mathcal{SR}}_{\mathcal{M}}(s)}}{\arg\max} \{r_{\omega_3}(s,s') + \gamma Q_\theta(s',s'')\},$$

---

**Algorithm 2** StaCQ (One Step RL version)

---

 1: **Input:** Dataset $\mathcal{D}$, $T$ number of iterations, $\tau$
 2: **Initialise:** parameters $\omega_1$, $\omega_2$, $\omega_3$, $\theta$, $\theta'$, $\phi$
 3: Pre-train $f_{\omega_1}$ and $I_{\omega_2}$ using Eqs.(7) and (8)
 4: Pre-train reachability criteria $\widehat{\mathcal{SR}}_{\mathcal{M}}$
 5: **for** $t = 1, \ldots, T$ **do**
 6:     Optimise reward function according to Eq. (11)
 7:     Optimise QSS-value according to Eq. (19)
 8:     Update target networks: $\theta' \leftarrow \tau\theta + (1-\tau)\theta'$.
 9: **end for**
10: **for** $t = 1, \ldots, T$ **do**
11:     Optimise policy: Eq. (20)
12: **end for**

---

Again, due to the forward model being fixed, a small amount of Gaussian noise is added to the policy action before being input into the model. This creates a more robust policy by ensuring the policy does not exploit the forward model. The complete procedure for the one step version of StaCQ is provided in Algorithm 2.

### D.3 One step method implementation details

In the original OneStepRL paper (Brandfonbrener et al., 2021), they evaluate their method over 10 seeds where the $\lambda$ hyperparameter has been tuned over the first 3 seeds and then evaluated with the $\lambda$ fixed for the remaining 7. However as we want a consistent hyperparameter for each environment we choose $\alpha = \{0.1, 1.0, 5.0\}$ for Hopper, Walker2d and Halfcheetah respectively and for the Antmaze tasks we choose $\alpha = \{10, 40, 100\}$ for the umaze, medium and large environments respectively. The one step method also uses a single critic, we found that increasing the number of critics deteriorated results. All other implementation details are the same as StaCQ, Appendix C.

## E Consistent hyperparameter experiment

The single StaCQ hyperparameter, $\lambda$ in the policy update, is tuned per environment (for example, using the same $\lambda$ for all Hopper datasets). This ensures StaCQ is not over-tuned to each dataset while still producing realistic, high-quality results. Most state-of-the-art methods have different hyperparameters for each dataset (Yu et al., 2020; Kidambi et al., 2020; Brandfonbrener et al., 2021; Wang et al., 2021; Ghasemipour et al., 2022; Rigter et al., 2022; Xu et al., 2022); this is not the case for StaCQ. However, TD3+BC and IQL both use a consistent hyperparameter across all datasets. TD3+BC uses an ensemble of two critics, whereas StaCQ has an ensemble of four. To compare fairly with these methods, we perform two comparisons. First, we fix StaCQ's hyperparameter across all locomotion tasks. Second, we tune TD3+BC's hyperparameter for each environment and supplement it with an ensemble of four critics. Table 2 shows these comparisons.

From Table 2, we can see that with consistent hyperparameter tuning strategies StaCQ still vastly outperforms TD3+BC (and IQL). This now provides a clear comparison showing how more Q-value evaluations through the state reachability metric allows for higher-quality policies than the batch-constrained counterparts.

| | | | Fixed hyperparameter | | Per environment tuning & 4 critics | |
| | | IQL | TD3+BC | StaCQ | TD3+BC | StaCQ |
|---|---|---|---|---|---|---|
| Random | Halfcheetah | - | 11.0 | **15.8** | $18.0 \pm 1.1$ | **$24.3 \pm 1.3$** |
| | Hopper | - | **8.5** | **8.6** | $7.6 \pm 0.7$ | **$17.5 \pm 12.8$** |
| | Walker2d | - | 1.8 | **3.5** | $0.9 \pm 0.2$ | **$4.4 \pm 6.4$** |
| Med | Halfcheetah | 47.4 | 48.3 | **50.7** | $49.0 \pm 0.3$ | **$57.6 \pm 0.6$** |
| | Hopper | 66.3 | 59.3 | **100.1** | $61.5 \pm 7.7$ | **$100.2 \pm 3.0$** |
| | Walker2d | 78.3 | 83.7 | **88.6** | $83.6 \pm 4.8$ | **$92.2 \pm 3.0$** |
| Med-Rep | Halfcheetah | 44.2 | **44.6** | **46.5** | $44.6 \pm 0.7$ | **$52.2 \pm 0.5$** |
| | Hopper | 94.7 | 60.9 | **98.7** | $61.5 \pm 23.8$ | **$99.1 \pm 1.3$** |
| | Walker2d | 73.9 | **81.8** | **85.5** | **$84.0 \pm 3.0$** | **$87.2 \pm 7.3$** |
| Med-Exp | Halfcheetah | 86.7 | **90.7** | **92.4** | $90.9 \pm 2.2$ | **$96.4 \pm 2.9$** |
| | Hopper | 91.5 | **98.0** | **94.0** | $104.5 \pm 11.8$ | **$111.9 \pm 0.2$** |
| | Walker2d | **109.6** | **110.1** | **114.2** | **$110.6 \pm 0.5$** | **$116.2 \pm 1.5$** |
| | Total | 692.6 | 698.7 | **798.6** | 716.7 | **859.2** |

Table 2: Average normalised scores for StaCQ, TD3+BC and IQL with consistent hyperparameter tuning strategies. The left hand side compares StaCQ with IQL and TD3+BC where there is a consistent hyperparameter across all datasets. The right hand side compares StaCQ with TD3+BC with 4 critics and hyperparameters tuned per dataset. The bold scores are within the 95% of the highest performing method in each dataset. Scores represent mean over 5 seeds of 10 evaluation $\pm$ the standard deviation.

# F Ablation study: exploring the effect of the reachability metric

In this section, we explore the effect of changing the reachability metric for StaCQ. That is, for fixed datasets, Walker2d-Medium and Walker2d-MediumExpert, we explore how the change of reachability norm and threshold for reachable states, $\epsilon$, effects the average normalised score.

Table 3 shows the comparison of the reachability metric with different thresholds. For the $L^1$-norm with $\epsilon = 0.01$ and $0.05$ and $l^\infty$-norm with $\epsilon = 0.01$ on the Walker2d-Medium dataset, the threshold was too small to find any reachable states. In these cases, the reachable states were the explicit next state in the dataset, therefore Eq. (10) and (12) where only updated using $(s, s') \in \mathcal{D}$. This is essentially the same method as TD3+BC, except using $Q(s, s')$ instead of $Q(s, a)$. In these cases, StaCQ performs on par with TD3+BC, as expected, but much worse than StaCQ with a larger threshold.

Otherwise, for all metrics increasing the threshold leads to an increased score up to a certain point (e.g. $\epsilon = 0.1$ for $L^2$-norm) where too many states are considered reachable, when they are not, at which point the average score decreases. On the Walker2d-medium dataset, StaCQ seems fairly robust to a change in reachability as even with the large thresholds the scores do not decrease as far as TD3+BC. On the Walker2d-MediumExpert dataset, when the threshold is too large, $\epsilon = 0.5$ for the $L^2$ and $L^\infty$ -norms, the performance degrades massively. This is because too many states are considered reachable which are in fact not.

As a result of this study, the reachability metric chosen across all datasets is the $L^\infty$-norm with $\epsilon = 0.1$, as these values produced the most reliable scores. It should be noted that other norms and thresholds could have been used to achieve similar or greater scores, however across all datasets these hyperparameters performed well.

| Norm | Threshold ($\epsilon$) | Walker2d-Medium | Walker2d-MediumExpert |
|---|---|---|---|
| $L^2$-norm | 0.01 | $94.1 \pm 3.1$ | $116.0 \pm 1.8$ |
| | 0.05 | $94.4 \pm 1.4$ | $104.5 \pm 0.1$ |
| | 0.1 | $92.1 \pm 1.0$ | $94.0 \pm 22.8$ |
| | 0.5 | $91.9 \pm 2.5$ | $0.3 \pm 0.6$ |
| $L^1$-norm | 0.01 | $82.6 \pm 1.3$ | $115.5 \pm 1.9$ |
| | 0.05 | $82.6 \pm 1.3$ | $114.6 \pm 1.8$ |
| | 0.1 | $84.1 \pm 17.0$ | $113.7 \pm 5.0$ |
| | 0.5 | $90.3 \pm 2.5$ | $107.8 \pm 11.1$ |
| $L^\infty$-norm | 0.01 | $82.6 \pm 1.3$ | $115.7 \pm 1.3$ |
| | 0.05 | $92.6 \pm 3.6$ | $114.2 \pm 1.9$ |
| | 0.1 | $92.2 \pm 3.0$ | $116.2 \pm 1.5$ |
| | 0.5 | $91.4 \pm 5.8$ | $6.7 \pm 12.4$ |

Table 3: Comparison of different reachability metric norms and thresholds on the Walker2d-Medium and Walker2d-MediumExpert datasets. The normalised score is averaged for 10 evaluations providing a standard deviation over 5 seeds.

## G   Learning curves for StaCQ

In this section we provide the learning curves of StaCQ, across all datasets. During training, at 5000 gradient step intervals, the mean of 10 evaluations (for the locomotion tasks) or 100 evaluations (for the Antmaze tasks) of StaCQ had been recorded across 5 seeds. This is used to monitor the performance of StaCQ during training and the environment interactions are not used for the policy or Q-value updates.

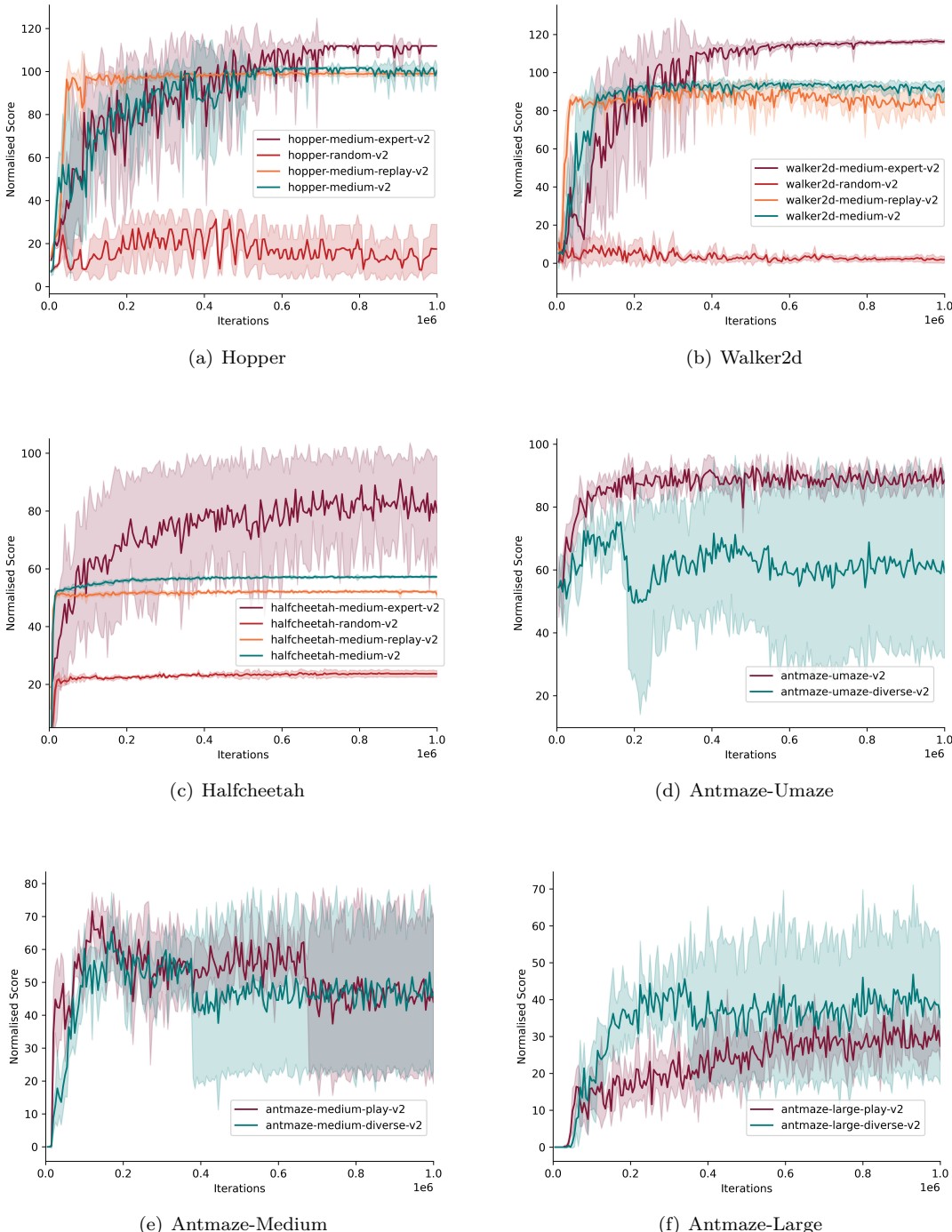

Figure 3: Learning curves for StaCQ across all datasets, grouped together by environment. The mean normalised score, over 10 evaluations (for locomotion tasks) or 100 evaluations (for Antmaze tasks), is recorded for every 5000 steps during training. The solid line is the mean score across 5 seeds and the shaded region is the standard deviation.

# H  Multi-step reachability: potential extensions and an illustrative example

The SCQL framework leverages one-step state reachability. A natural and promising direction for future work involves extending this concept to *multi-step reachability*. The core idea is to use a learned forward dynamics model, $f_{\omega_1}$, to simulate $k$-step trajectories $(s_0, a_0, \ldots, a_{k-1}, s_k)$ starting from a state $s_0$. The crucial constraint remains that the trajectory must *terminate* at a state $s_k$ that belongs to the original dataset $\mathcal{D}$. This allows the agent to potentially bridge larger gaps in the dataset by traversing through intermediate states $(s_1, \ldots, s_{k-1})$ that might be out-of-distribution, while still anchoring the final value estimate to reliable in-distribution data ($s_k \in \mathcal{D}$).

**The two-step case**  Let us examine the specific case of two-step reachability. We define the set of 2-step reachable states ending in the dataset $\mathcal{D}$ from a state $s$ as:

$$\mathcal{SR}_{\mathcal{M},\mathcal{D}}^{(2)}(s) = \{s_2 \in \mathcal{D} \mid \exists a_0, a_1 \text{ s.t. } p(s_1|s, a_0) = 1 \text{ and } p(s_2|s_1, a_1) = 1\}.$$

Here, the intermediate state $s_1$ does not need to be in $\mathcal{D}$. Practically, reachability would be estimated using the learned model $f_{\omega_1}$, denoted $\widehat{\mathcal{SR}}_{\mathcal{M},\mathcal{D}}^{(2)}(s)$.

To utilise this, a modified Q-learning update could estimate the value $Q(s, s')$ based on the best 2-step reachable state from $s'$. A conceptual sketch, analogous to Eq. (5), might look like:

$$Q(s, s') \leftarrow (1 - \alpha)Q(s, s') + \alpha \left[ r(s, s') + \gamma \max_{\substack{s''' \in \mathcal{D} \\ \cap \\ s''' \in \widehat{\mathcal{SR}}_{\mathcal{M},\mathcal{D}}^{(2)}(s')}} V^{(2)}(s', s''') \right].$$

In this sketch, $V^{(2)}(s', s''')$ represents the estimated value of the optimal 2-step path initiating from $s'$ and terminating at $s''' \in \mathcal{D}$. Accurately estimating $V^{(2)}$ using the learned model $f_{\omega_1}$ (involving rewards for the first step $s' \to s''$ and the value/reward for the second step $s'' \to s'''$) is a key challenge, requiring careful handling of model predictions and potential accumulated errors.

Extracting an optimal policy $\pi^*(s)$ based on $k$-step reachability is also more complex than the 1-step case. Conceptually, the goal is to select the first action $a_0$ that initiates the sequence leading to the highest-value $k$-step reachable state $s_k^* \in \mathcal{D}$:

1. Find the best $k$-step endpoint: $s_k^* = \arg\max_{s_k \in \widehat{\mathcal{SR}}_{\mathcal{M},\mathcal{D}}^{(k)}(s)} V^{(k)}(s, s_k)$.
2. Determine the first action $a_0^*$ on the optimal path from $s$ to $s_k^*$.

Step 2 typically involves some form of planning (e.g., sampling action sequences with the model) or learning a policy specifically optimised for this multi-step objective.

**Illustration: maze two-step reachability**  The potential benefit of multi-step reachability is illustrated in the simple maze environment shown in Figure 4. The dataset provided (Figure 4a) is sparse, preventing the standard one-step SCQL from finding a path to the goal from all dataset states (Figure 4b). However, by considering two-step reachability ($k = 2$), the agent can plan through an intermediate state that may not be in the dataset (indicated by dotted arrows). Assuming perfect knowledge of the dynamics for this illustrative example, a hypothetical two-step state-constrained policy can successfully connect all starting points to the goal region (Figure 4c), demonstrating improved trajectory stitching capabilities.

**Potential challenges**  While promising, extending state-constrained RL to multiple steps also introduces significant practical challenges, primarily stemming from reliance on the learned dynamics model $f_{\omega_1}$. Key challenges and potential research directions include:

- **Model error accumulation:** Errors in $f_{\omega_1}$ compound over longer rollouts ($k > 1$), leading to inaccurate state predictions and potentially unreliable value estimates for the $k$-step paths.

- **Handling uncertainty:** Effectively leveraging multi-step rollouts requires managing this model uncertainty. Potential approaches include incorporating pessimism into value estimates based on

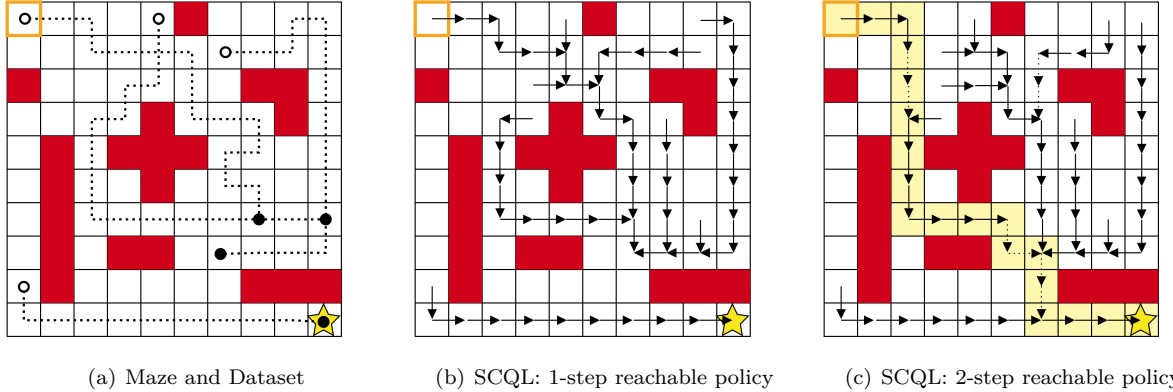

(a) Maze and Dataset  (b) SCQL: 1-step reachable policy  (c) SCQL: 2-step reachable policy

Figure 4: Comparison of one-step and two-step state reachability SCQL methods on a simple maze environment. (a) The maze is a 10 by 10 grid where the coordinate values $(x, y)$ represent the state. The high reward region is represented by a star and maze walls are represented by red grid squares. The dataset is composed of 4 trajectories represented by the dotted lines where the white circle is the starting state and the black circle is the final state. (b) The final policy when applying one-step reachable SCQL to the dataset. (c) The final policy when applying two-step reachable SCQL to the dataset, where dotted arrows represent transitions through a state that is not in the dataset.

rollout uncertainty (Yu et al., 2020; Rigter et al., 2022), using model ensembles to quantify confidence, or dynamically adapting the rollout horizon $k$ based on model reliability.

- **Computational cost:** Finding and evaluating all $k$-step reachable paths terminating in $\mathcal{D}$ can be computationally demanding. Efficient search or sampling techniques would be crucial. Graph-based search algorithms operating on the state space defined by the learned model could offer an alternative to simple rollouts for finding paths back to $\mathcal{D}$.

- **Algorithmic integration:** Developing robust Q-learning or policy optimization methods that correctly incorporate these multi-step, state-constrained value estimates is non-trivial, especially when considering the complexities of planning or differentiating through model rollouts.

Exploring these multi-step extensions involves balancing the benefit of connecting more distant states against the cost and unreliability of longer model-based rollouts. This intersection of model-based planning and offline constraints presents a rich area for future investigation within the state-constrained RL paradigm.

