# OpenReview forum: "State-Constrained Offline Reinforcement Learning"
_TMLR — Accepted by TMLR_

### Review · Reviewer_vVkQ · 2024-12-09

**Summary Of Contributions:**

This paper discusses offline reinforcement learning with the additional constraint that the states need to exist in the training dataset. Existing theoretical results by Fujimoto et al. are extended to the proposed state-constrained QSS-learning (SCQL) approach and a practical implementation of SCQL (called StaCQ) is evaluated on the D4RL environments for offline RL and compared with model-free baselines (BC, BCQ, TD3+BC, IQL) as well as model-based baselines (MBTS, Diffuser, RAMBO). A variant of their implementation that learns the values and policies together (called OneStep StaCQ) is also discussed and evaluated.

**Audience:**

Yes

**Broader Impact Concerns:**

No concerns.

**Claims And Evidence:**

Yes

**Requested Changes:**

- The claims of this work should be stated in a more clearer way. I suggest the authors add a section that clearly states the claims/contributions made in this work.
- Formatting of the references can be improved at some points (e.g., “q(s, s’)”, “D4rl”, “offline rl”, “q-learning”)
- Minor comments regarding notation:
    - \argmax should be properly defined. Please, do not use \arg \max.
    - In Eq. (5) and (6) the subscripts to the max could be improved: Either write this as a intersection of the two sets $\mathcal{D}$ and $\mathcal{SR}_\mathcal{M}(s)$ or add a logical and-operator.

**Strengths And Weaknesses:**

Strengths:

- The proposed approach is interesting and worth studying.
- The paper is well-written and explains necessary background (QSA-learning and QSS-learning) at an appropriate level of detail.
- The paper contains theoretical and experimental results showing the efficacy of the proposed approach and the implementation.
- The mathematical notation is clean and understandable.

Weaknesses:

- Overall, this is, in my opinion, solid work and suitable for publication in TMLR. There are only minor weaknesses, which I list in the requested changes section of my review.

---

> ### Author Response · Authors · 2025-03-19
> **Reply to Reviewer vVkQ**
>
> We would like to thank the reviewer for their constructive feedback and encouraging evaluation of our work. We have taken care to address the requested changes, which are summarized below:
>
> 1. **Clearer Claims and Contributions**
>
>     To ensure our claims and contributions are clearly stated, we have revised the final paragraph of the introduction as follows:
>
>     In this work, we make several key contributions. First, building on the concept of reachability, we introduce state-constrained offline RL. Unlike batch-constrained RL, which anchors learning to specific state–action pairs, state-constrained RL focuses on states, allowing more flexible action selection while still mitigating distributional shift. We also provide theoretical convergence guarantees under deterministic assumptions, showing that our framework yields policies whose actions are of higher or equal value compared to batch-constrained approaches. Another main contribution is StaCQ, a novel deep learning algorithm that learns a state-constrained value function and updates the policy to stay close to the highest-quality reachable states. We demonstrate competitive performance on multiple D4RL tasks, surpassing many state-of-the-art methods in locomotion and Antmaze. This positions StaCQ as a robust baseline for future work on state-constrained RL, similarly to how BCQ (Fujimoto et al., 2019) has served as a strong baseline for batch-constrained techniques, thereby advancing offline RL research
>
> 2. **Reference formatting**
>
>     We have improved our references to ensure consistent formatting (e.g., “D4RL,” “offline RL,” “Q-learning”).
>
> 3. **Notation Improvements**
>
>     We have properly defined \argmax, and the subscripts in Eqs. (5) and (6) are now written as intersections of the relevant sets.
>
>
> We hope these revisions address all of your concerns and greatly appreciate your helpful review, which has contributed to strengthening our manuscript. Thank you again for your time and thoughtful comments.

---

### Review · Reviewer_xzyQ · 2024-12-15

**Summary Of Contributions:**

This paper introduces a novel algorithm constraining the policy to produce actions that lead to reachable states within the datasets. While popular batch-constrained methods confine the policy to the state-action pairs from the dataset, this work proposed constraint allows for state-action pairs that do not exist in the datasets but are reasoned from the dataset. The optimality related to the proposed method is proofed, and experiments on the D4RL benchmark present better performance than the selected baselines.

**Audience:**

Yes

**Broader Impact Concerns:**

I believe that this work does not raise concerns regarding Broader Impact

**Claims And Evidence:**

No

**Requested Changes:**

- A performance comparison between agents using a consistent hyperparameter tuning (see weakness 1) is needed. Either using domain-specific hyperparameters or similar values for all environments.

- A discussion about the StaCQ components (See weakness 2) and their influence on the performance is needed.

- A missing definition of $s_{terminal}$ (Definition 3.3). Is $s_{terminal}$ used in this work similar to the definition from Fujimoto et al., [1]? Please define this notation in this work.

- Could you provide the environment penalty and reward values of the maze environment (Section 3.2). Currently, the paper only mentions *"a small negative value"* and *"rewarded with a large positive value"*. Providing these values and the training details (e.g. agent hyperparameters) will help the readers further understand the experiment.

### Minors

- In the Introduction section, *"However, adhering strictly to the state-action distribution can be restrictive. Thus, recent techniques..."* The first sentence makes a claim that is not discussed. A brief discussion about "can be restrictive" is needed here.

- Section 3.2, *"With each movement, the agent incurs a reward penalty of a small negative value, but upon..."*, what is the "reward penalty"? Here, it should be a penalty, given the context.

- Consistency of using "co-ordinate" and "coordinate"; "black circle" "white ones" and "closed circle" "open circle"

- The notation $Q(s, \pi(s))$ or $f(s, \pi(s))$ implies a deterministic policy $\pi(s)$. Although this paper considers deterministic policy in their experiments, please explicitly define the deterministic policy $\pi(s)$ in Section 4.2.

## Reference
[1] Scott Fujimoto, David Meger, and Doina Precup. Off-policy deep reinforcement learning without exploration. In International conference on machine learning, pp. 2052–2062. PMLR, 2019.

**Strengths And Weaknesses:**

# Strengths
- The paper is well-written. I enjoyed my reading of this work.
- The proposed method is novel and intuitive. Constraining on reachable states allows more action generalization than the batch constraints.
- The illustrative experiment presents the motivation of the SCQL.
- The proven convergence and optimality properties allow the comparison of the SCQL and BCQ, showing the theoretical advantage of SCQL.

# Weaknesses
- The implemented agent of SCQL applies ensembles of forward, inverse, and critic models. In addition, the implementation of the Locomotion task is different from that of the Antmaze task in using a shared/independent target network, and whether to use a reward model. Although using these techniques does not count for weaknesses, their effect on performance is not discussed in this paper, which makes it vague whether the proposed method itself is the main source of the presented performance improvement (Table 1)

- The comparison shown in Table 1 uses different hyperparameter tuning strategies for each agent. For example, the TD3-BC (Fujimoto et al.[1]) and IQL (Kostrikov et al.[2]) are taken from the original papers, where similar hyperparameter values are used for all Locomotion tasks. However, StaCQ uses domain-specific values for its hyperparameters. Some baseline agents might perform better with their parameters tuned for each environment. Based on this, Table 1 does not present a clear comparison.

## Reference
[1] Fujimoto, Scott, and Shixiang Shane Gu. "A minimalist approach to offline reinforcement learning." Advances in neural information processing systems 34 (2021): 20132-20145.
[2] Kostrikov, Ilya, Ashvin Nair, and Sergey Levine. "Offline Reinforcement Learning with Implicit Q-Learning." International Conference on Learning Representations.

---

> ### Author Response · Authors · 2025-03-19
> **Reply to Reviewer xzyQ (1/2)**
>
> We would like to thank the reviewer for their insightful comments and for highlighting points that helped improve our manuscript. Below, we address the weaknesses and requested changes:
>
> **Weaknesses**
>
> 1. **Use of ensembles and difference in implementation (Antmaze vs. Locomotion)**
> We have added a discussion in Sections 6 and 7 to clarify these design choices. Our forward, and inverse models are integral to the state reachability estimation and not subject to ablation, as they jointly estimate environment dynamics. We adopt ensembles to reduce model error [1–4], balancing accuracy with computational cost. We use an ensemble of 4 critics for StaCQ and 1 critic for the one-step StaCQ method, these are chosen due to stability while staying small as possible. We switched to independent target networks for Antmaze (inspired by Ghasemipour et al. [5]) to improve performance in medium and large mazes. A reward model is unnecessary in Antmaze because its sparse reward is fully specified by the next state. We now highlight these points in Section 6.
> 2. **Hyperparameter tuning in Table 1 comparison**
> StaCQ uses a single hyperparameter per environment (not tuned for dataset quality), while other methods often tune hyperparameters per dataset [2,5–11]. Despite differences in how these hyperparameters scale (e.g., actions in [-1,1] vs. unbounded states), we agree that a direct comparison with TD3+BC [12] benefits clarity. We have therefore fixed StaCQ’s hyperparameter and re-tuned TD3+BC (with four critics) using a matching strategy. Please see Appendix E for a new table comparing both methods under consistent tuning. Even with this adjustment, StaCQ maintains superior performance on most tasks.
>
> **Requested Changes**
>
> 1. **Performance comparison with consistent hyperparameter tuning**
> Addressed in Appendix E (see updated table).
> 2. **Discussion of StaCQ components**
> Included in Sections 6 and 7 (see above).
> 3. **Definition of $s_{\text{terminal}}**
> We confirm it follows Fujimoto et al. (2019) [13] and have clarified this in Definition 3.3.
> 4. **Environment penalties and reward values in Section 3.2**
> We now provide full training details: $\alpha=0.25, \gamma=0.99, N=100, r_{\text{pen}}=−0.1$, and $r_{\text{goal}}=10$.
>
> **Minor Points**
>
> - Revised “*However, adhering strictly to the state-action distribution can be restrictive when the optimal action is not found in the dataset*” to clarify how this restricts actions not present in the dataset.
> - Clarified that “reward penalty” in Section 3.2 is a small negative reward.
> - Changed references to “coordinate,” “white circle,” and “black circle.”
> - Added the deterministic policy definition $\pi:\mathcal{S}\rightarrow \mathcal{A}$ in Section 4.2.
>
> We appreciate the reviewer’s feedback and believe these revisions address all concerns. Thank you again for helping us improve the manuscript.

---

> > ### Author Response · Authors · 2025-03-19
> > **Reply to Reviewer xzyQ (2/2)**
> >
> > **References**
> >
> > [1] Michael Janner, Justin Fu, Marvin Zhang, and Sergey Levine. When to trust your model: Model-based policy optimization. Advances in neural information processing systems, 32, 2019
> >
> > [2]   Arthur Argenson and Gabriel Dulac-Arnold. Model-based offline planning. arXiv preprint arXiv:2008.05556, 2020.
> >
> > [3] Jacob Buckman, Danijar Hafner, George Tucker, Eugene Brevdo, and Honglak Lee. Sample-efficient reinforcement learning with stochastic ensemble value expansion. Advances in neural information processing systems, 31, 2018.
> >
> > [4] Kurtland Chua, Roberto Calandra, Rowan McAllister, and Sergey Levine. Deep reinforcement learning in a handful of trials using probabilistic dynamics models. Advances in neural information processing systems,31, 2018.
> >
> > [5] Kamyar Ghasemipour, Shixiang Shane Gu, and Ofir Nachum. Why so pessimistic? estimating uncertainties for offline rl through ensembles, and why their independence matters. Advances in Neural Information Processing Systems, 35:18267–18281, 2022.
> >
> > [6] David Brandfonbrener, Will Whitney, Rajesh Ranganath, and Joan Bruna. Offline rl without off-policy evaluation. Advances in neural information processing systems, 34:4933–4946, 2021.
> >
> > [7] Rahul Kidambi, Aravind Rajeswaran, Praneeth Netrapalli, and Thorsten Joachims. Morel: Model-based offline reinforcement learning. Advances in neural information processing systems, 33:21810–21823, 2020.
> >
> > [8] Marc Rigter, Bruno Lacerda, and Nick Hawes. Rambo-rl: Robust adversarial model-based offline reinforcement learning. Advances in neural information processing systems, 35:16082–16097, 2022
> >
> > [9]  Jianhao Wang, Wenzhe Li, Haozhe Jiang, Guangxiang Zhu, Siyuan Li, and Chongjie Zhang. Offline reinforcement learning with reverse model-based imagination. Advances in Neural Information Processing Systems, 34:29420–29432, 2021.
> >
> > [10] Haoran Xu, Li Jiang, Li Jianxiong, and Xianyuan Zhan. A policy-guided imitation approach for offline reinforcement learning. Advances in Neural Information Processing Systems, 35:4085–4098, 2022.
> >
> > [11] Tianhe Yu, Garrett Thomas, Lantao Yu, Stefano Ermon, James Y Zou, Sergey Levine, Chelsea Finn, and Tengyu Ma. Mopo: Model-based offline policy optimization. Advances in Neural Information Processing Systems, 33:14129–14142, 2020.
> >
> > [12] Fujimoto, Scott, and Shixiang Shane Gu. "A minimalist approach to offline reinforcement learning." Advances in neural information processing systems 34 (2021): 20132-20145.
> >
> > [13] Scott Fujimoto, David Meger, and Doina Precup. Off-policy deep reinforcement learning without exploration. In International conference on machine learning, pp. 2052–2062. PMLR, 2019.

---

### Review · Reviewer_XLFq · 2025-03-11

**Summary Of Contributions:**

This paper presents a simple yet effective idea—QSS learning operates within a restricted state-reachable domain.

**Audience:**

Yes

**Claims And Evidence:**

Yes

**Requested Changes:**

•	Extend the theoretical analysis beyond deterministic MDPs and provide quantitative learning efficiency guarantees.

•	Present key theoretical results using formal mathematical expressions.

•	Modify the maze navigation experiment to ensure that the state-reachable domain is learned rather than manually provided.

**Strengths And Weaknesses:**

**Strengths:**

•	The proposed idea is conceptually straightforward.

•	The method effectively constrains learning within a reachable state space, which may improve robustness in offline RL.

**Weaknesses:**

1.	**Theoretical Limitations:**

•	The theoretical analysis is confined to deterministic MDPs, which limits its applicability to broader RL settings.

•	The provided analysis only establishes the possibility of convergence but lacks a quantitative learning efficiency analysis. This makes the theoretical contribution more of a descriptive validation of QSS learning rather than a comparative study with traditional QSA learning. A theoretical comparison, particularly in terms of convergence rate and sample complexity, would strengthen the contribution.

2.	**Writing and Mathematical Rigor:**

•	The key theorems are stated in natural language rather than using formal mathematical expressions. To improve clarity and rigor, I recommend expressing these results mathematically.

3.	**Maze Navigation Experiment Issues:**

•	The direct given of the state-reachable domain in the maze navigation task appears to be a form of implicit cheating, as it directly provides environment information rather than learning it from data.

•	This approach may also introduce errors, as it does not consider environmental constraints such as walls blocking the agent’s movement.

•	A fairer comparison should involve learning the state-reachable domain from the dataset rather than assuming it a priori.

---

> ### Author Response · Authors · 2025-03-19
> **Reply to Reviewer XLFq**
>
> We appreciate your careful reading and the constructive feedback. We have revised our manuscript accordingly and address each concern below:
> ### **1. Theoretical limitations**
> We acknowledge that our theoretical analysis focuses on deterministic MDPs. This choice follows prior work such as BCQL [1], which also adopts deterministic transitions to establish convergence properties. Deterministic MDPs often serve as a clear first step, as they simplify discussion and allow direct comparisons of SCQL (our method) with batch-constrained approaches. Extending these results to stochastic MDPs would require redefining state reachability under probabilistic transitions, an important avenue for future work.
>
> Our primary theoretical goal here is to demonstrate convergence and compare SCQL with BCQL, rather than provide a comprehensive analysis of convergence rates or sample complexity. While these quantitative metrics are undoubtedly valuable, exploring them would involve considerably more complex assumptions and derivations that could overshadow our main focus. As such, our current results—both theoretical and empirical—provide a foundational perspective on state-constrained RL. We have expanded our discussion in the revised manuscript to clarify this point, and Theorem 3.7 in particular highlights SCQL's advantages over BCQL when the state reachability function is known.
> ### **2. Writing and mathematical rigour**
> In response to your request, we have strengthened the presentation of Theorems 3.5, 3.6, and 3.7 by incorporating clearer mathematical statements:
>
> **Theorem 3.5:** Under assumptions A1-4 and assuming every state $s$ is encountered infinitely, let $Q_n(s,s')$ be the value from the $n$th update of Eq.(5), the update rule of SCQL,  then $Q_n(s,s')$ converges to the optimal QSS-value $Q^*(s,s')$, as $n \rightarrow \infty$ for all $(s,s')$.
>
> **Theorem 3.6:** Under assumptions A1-4 and assuming every state $s$ is encountered infinitely, let $Q_n(s,s')$ be the value from the $n$th update of Eq.(5), the update rule of SCQL,  then $Q_n(s,s')$ converges to $Q_{\mathcal{S}}^{\pi}(s,s')$, the optimal QSS-value computed from states from $\mathcal{D}$, with the optimal state-constrained policy defined by Eq. (6) where $s \in \mathcal{D}$ and $s' \in \mathcal{SR}_{\mathcal{M}}(s) \cap \mathcal{D}$.
>
> **Theorem 3.7.** Let $\pi_{\text{BCQL}}$ and $\pi_{\text{SCQL}}$ be the policies produced by BCQL and SCQL respectively. Then, in a deterministic MDP, $V_{\pi_{\text{SCQL}}}(s) \geq V_{\pi_{\text{BCQL}}}(s)$,  $\forall s \in \mathcal{D}$.
>
> All other theorems match the language presented in BCQL [1].
> ### **3. Maze navigation experiment**
> We would like to clarify the motivation for the example in Section 3.2, as it may not have been fully clear. It is intended as an illustrative “best-case” scenario to show how SCQL leverages  state reachability to outperform BCQL. This serves as a pedagogical tool to visually demonstrate the effectiveness of leveraging state reachability, and it is deliberately designed to highlight conditions under which SCQL excels.
>
> We emphasise that walls and other constraints are not ignored. Since such states do not appear in the dataset, they are automatically excluded from SCQL’s updates. Even when specifying the state-reachable domain, the agent cannot transition through walls if these states are absent from the dataset. In the revised manuscript, we have now clarified these details and further highlighted that this example is intended to illustrate our method’s underlying principle and how it differs from BCQL.
>
> For scenarios in which state reachability must be learned, we refer the reader to Section 6. There, the reachability function is estimated rather than given. This reflects real-world offline RL settings and shows that SCQL remains effective without a priori state-transition knowledge.
>
> We hope these clarifications and revisions address your concerns. We value your feedback as it has helped strengthen both the theoretical and empirical aspects of our work.
>
> ### References
>
> [1] Scott Fujimoto, David Meger, and Doina Precup. Off-policy deep reinforcement learning without exploration. In International conference on machine learning, pp. 2052–2062. PMLR, 2019.

---

### Review · Reviewer_72cw · 2025-03-25

**Summary Of Contributions:**

The paper introduces a formal state-constrained offline RL framework grounded in new theoretical insights. It defines state reachability (Definition 3.1) as the set of next states obtainable from a given state via some action in a deterministic MDP. This notion allows the Q-function to consider multiple possible next states per state, rather than a single observed state-action pair. Based on this, the authors formulate QSS-learning, where the Q-value is defined over state-to-state transitions $Q(s,s’)$ instead of state-action pairs. Under standard assumptions ((A1)-(A4): deterministic environment, bounded rewards, finite initial Q-values, $0\le\gamma<1$), they prove that this QSS-learning converges to the optimal state-to-state value function (Theorem 3.2) by a straightforward extension of classic Q-learning proofs. The theoretical development is rigorous, adapting known results (e.g. convergence of Q-learning in deterministic MDPs) to the new setting with appropriate justification. Proofs are provided (many deferred to the appendix for brevity), and they appear logically sound given the assumptions.

**Audience:**

Yes

**Broader Impact Concerns:**

No Broader Impact Concerns.

**Claims And Evidence:**

Yes

**Requested Changes:**

Please answer the questions for the weakness part in above.

**Strengths And Weaknesses:**

**Strengths:** This work proposes a new paradigm for offline RL: focusing on the state distribution of the dataset rather than the state-action distribution. This state-constrained approach is a departure from nearly all existing offline RL methods, which are batch-constrained (i.e. they confine the policy or value updates to logged $(s,a)$ pairs). In contrast, the authors allow the agent to consider out-of-distribution actions as long as those actions lead to in-distribution states. This is a novel relaxation of the offline RL constraints – essentially enabling the agent to explore unseen actions safely by keeping it within the support of seen states. Prior methods either strictly prohibit unseen actions (to avoid distribution shift) or heavily penalize them, which can prevent the agent from improving beyond the behavior policy. By confining only to states, the proposed framework permits the policy to reach new combinations of state transitions that were never directly observed, as long as those states themselves are known. This idea has direct implications for trajectory stitching: the algorithm can piece together fragments of suboptimal trajectories from the dataset into a better trajectory that achieves higher reward. Rather than being limited to following the dataset action-by-action, the agent can stitch via adjacent states – effectively jumping from one trajectory to another at a common or nearby state. This capability is a unique contribution and addresses a known challenge in offline RL, where improving beyond the given trajectories is hard. The paper illustrates this advantage in concept with Figure 1: the state-constrained method identifies high-value reachable states (yellow circles) that were not directly connected in the dataset and uses them to improve the policy. This is a clear innovation over batch-constrained methods that “lock” the policy to recorded actions and cannot leverage such alternate transitions.

**Weakness:**

1. Assumption of Deterministic Dynamics: The theoretical framework and experimental setup assume deterministic environments, which simplifies the definition of reachability (i.e., an action deterministically leads to a specific next state). Most real-world environments have stochastic transitions due to noise, uncertainty, or partial observability. In such cases, defining and verifying whether a state is “reachable” becomes less clear.

2. One-Step Reachability Only: The method only considers one-step reachability, i.e., whether a state can be reached from another in a single action. This limits the ability to stitch longer sub-trajectories, potentially capping performance in long-horizon or sparse-reward settings.

3. Limited Exploration of Stochastic or Realistic Benchmarks: All experiments were done in relatively deterministic simulators (MuJoCo, AntMaze, grid-worlds). The generalizability to real-world applications or highly stochastic tasks is uncertain.

4. No Direct Safety Guarantees: Although StaCQ only considers actions that lead to known states, it does not provide formal guarantees about safety or worst-case performance in unknown situations.

---

> ### Author Response · Authors · 2025-03-28
> **Reply to Reviewer 72cw (1/2)**
>
> Thank you for your thorough review and positive assessment of our work's novelty and contributions. We appreciate your constructive feedback regarding the weaknesses. Below, we address each point:
>
> 1. **Assumption of deterministic dynamics**
>
>     You are correct that our current theoretical framework and experimental setup assume deterministic MDPs. We believe this focus is appropriate and sufficient for this initial work introducing the novel state-constrained paradigm for several reasons. Firstly, adopting the deterministic assumption allows for a clear and rigorous theoretical analysis, enabling direct comparison with the foundational theory of batch-constrained methods like BCQ, which also initially relied on this simplification. This was crucial for formally establishing the core theoretical benefits of our approach, such as convergence with potentially smaller dataset requirements (Theorem 3.6) and guaranteed performance improvement over BCQL under these conditions (Theorem 3.7). Secondly, it facilitates fair and interpretable experimental comparisons with closely related algorithms like TD3+BC, isolating the impact of the state-constrained approach itself. Establishing the fundamental concept and demonstrating its potential in this well-defined, albeit simplified, setting is a key contribution of this paper, providing a necessary foundation before tackling the additional complexities of stochastic environments.
>
>     While acknowledging the importance of stochasticity for real-world applicability, we view the extension to such settings as significant future work, building upon the theoretical and empirical groundwork laid here. As discussed in Section 7, potential avenues exist, such as defining reachability via transition likelihoods (e.g., $s' \in \mathcal{SR}(s)$ if $p(s'|s,a) = \max_{s' \sim p(\cdot|s,a)} p(s'_i|s,a)$ ). We have ensured this perspective is clear in our discussion.
>
> 2. **One-step reachability only:**
>
>     Our focus on one-step reachability is a deliberate choice for this foundational work. It represents the most fundamental setting, allowing us to clearly introduce, analyse, and validate the core concept of the state-constrained offline RL paradigm without the immediate complexities of multi-step transitions. We view this not as a weakness, but as the necessary starting point for this line of research.
>
>     We agree that extending this concept to multi-step reachability (e.g., k-step) is a natural and promising advancement that builds upon the foundation laid here, potentially enhancing capabilities like trajectory stitching over longer horizons, as noted in our discussion. However, this extension introduces significant challenges, notably the issue of compounding model errors during rollouts. Effectively addressing these errors requires dedicated investigation and likely constitutes a distinct, more advanced state-constrained method. Therefore, we consider multi-step reachability an important extension for future work, rather than a shortcoming of this initial paper.
>
>     To underscore that this is a planned direction and illustrate the potential, we have included an example in Appendix H showing how a 2-step reachability definition could be exploited, highlighting the flexibility and extensibility of the state-constrained framework presented.
>
> 3. **Limited exploration of stochastic or realistic benchmarks:**
>
>     We chose the D4RL benchmark suite for our experiments as it is the established standard in offline RL research. Using these widely accepted benchmarks, even though primarily deterministic, is crucial for ensuring direct, fair, and reproducible comparisons against numerous prior works, including the key baselines TD3+BC, IQL, and RAMBO.
>
>     While these MuJoCo and Antmaze environments are indeed simulators, they include challenging tasks recognized by the community. Notably, the Antmaze tasks feature sparse rewards, environments where StaCQ demonstrates competitive performance against model-free methods and outperforms RAMBO, indicating the approach's applicability beyond simple dense-reward settings.
>
>     As mentioned in point 1, extending and evaluating the state-constrained framework in explicitly stochastic environments remains an important avenue for future work.

---

> > ### Author Response · Authors · 2025-03-28
> > **Reply to Reviewer 72cw (2/2)**
> >
> > 4. **No direct safety guarantees:**
> >
> >     While our method does not offer formal safety guarantees in the sense of worst-case bounds under arbitrary distribution shift, it provides a practical safety benefit by design. By constraining updates and policy optimisation to transitions leading to states *known* to be in the dataset distribution ($s' \in \mathcal{D}$ and $s' \in \mathcal{SR}_{\mathcal{M}}(s)$), StaCQ inherently avoids executing actions that lead to entirely unknown regions of the state space, unlike unconstrained methods. This can be viewed as a form of safe exploration within the offline setting.
> >
> >     Theoretically, in the worst-case scenario where the estimated reachable states $\hat{\mathcal{SR}}_{\mathcal{M}}(s)$ only contain the explicitly observed next state $s'$ for every $s$ in the dataset, the state-constrained MDP $\mathcal{M_S}$ becomes equivalent to the batch-constrained MDP. Consequently, SCQL's performance would match that of BCQL, as shown in the proof of Theorem 3.7.
> >
> > We hope this addresses your concerns and clarifies the motivations and scope of our current work. We have incorporated clarifications related to stochasticity and multi-step reachability into the discussion (Section 7) and added the illustrative example in Appendix H.

---

### Decision · Action_Editor_XqhE · 2025-05-01

**Recommendation:** Accept as is

**Comment:**

As previously mentioned, most of the pushback on this paper was of the form 'please do more', which shouldn't count against its acceptance, as what is there is sound and relevant.

**Audience:**

While the current limitation to deterministic dynamics does somewhat limit its audience, the authors are right to point out that this is just an initial limitation and engagement with TMLR's broad audience could help in this regard. Offline RL is generally of broad appeal, so I'm confident TMLR's audience will be interested.

**Claims And Evidence:**

The author's theoretical and empirical contributions were, respectively, technically sound and experimentally valid. There was some concern around the depth of the theoretical analysis and the limited scope of the empirical work, but this sort of pushback can be applied to most work and shouldn't be taken as a knock against its fundamental soundness.